# ANOMALY DETECTION BY CONTEXT CONTRASTING

## ABSTRACT

Anomaly detection focuses on identifying samples that deviate from the norm. When working with high-dimensional data such as images, a crucial requirement for detecting anomalous patterns is learning lower-dimensional representations that capture concepts of normality. Recent advances in self-supervised learning have shown great promise in this regard. However, many successful self-supervised anomaly detection methods assume prior knowledge about anomalies to create synthetic outliers during training. Yet, in real-world applications, we often do not know what to expect from unseen data, and we can solely leverage knowledge about normal data. In this work, we propose $CON_2$, which learns representations through context augmentations that allow us to observe samples from two distinct perspectives while keeping the invariances of normal data. $CON_2$ learns rich representations of context-augmented samples by clustering them according to their context while simultaneously aligning their positions across clusters. At test time, representations of anomalies that do not adhere to the invariances of normal data then deviate from their respective context cluster. Learning representations in such a way thus allows us to detect anomalies without making assumptions about anomalous data.

## 1 INTRODUCTION

Reliably detecting anomalies is essential in many safety-critical fields such as healthcare (Schlegl et al., 2017; Ryser et al., 2022), finance (Golmohammadi & Zaiane, 2015), industrial fault detection (Atha & Jahanshahi, 2018; Zhao et al., 2019), or cyber-security (Xin et al., 2018). A common real-world example of anomaly detection is the standard screening scenario, where doctors regularly examine the general population for anomalies that would indicate a health risk. Standard screening datasets thus predominantly comprise samples from healthy people, as most screened individuals do not exhibit any diseases. Detecting anomalies in this setting is challenging, as anomalies can arise from an arbitrary set of potentially rare diseases or measurement errors, while we predominantly encounter normal samples from healthy people in the dataset. The field of anomaly detection tackles such problems by learning representations that reflect normality during training and, at test time, detecting anomalies as deviations from the learned normal structure (Ruff et al., 2021).

Recent works have demonstrated that learning a representation space containing features that tightly represent normality is essential for anomaly detection (Ruff et al., 2018; Oza & Patel, 2018; Sabokrou et al., 2020). Current state-of-the-art methods carefully design synthetic anomalies and explicitly encourage anomalous representations to be different from normal ones (Tack et al., 2020; Wang et al., 2023). However, anomalies can be diverse and unexpected, making it difficult to simulate them in real-world settings.

This work presents a novel anomaly detection objective, $CON_2$[1], which learns informative, tightly clustered representations of normal samples. We illustrate an overview of the algorithm in Figure 1. Unlike previous works, which focus on prior knowledge about anomalies, the proposed $CON_2$ models properties of normal samples, which is particularly useful in more specialized data, such as in the medical domain, which we demonstrate in our experiments. $CON_2$ leverages *context augmentations* that let us observe samples in different contexts while preserving their normal content. Our new $CON_2$ objective clusters representations according to these contexts while encouraging similar rep-

---

[1] The code is attached to this submission and will be made publicly available upon acceptance.

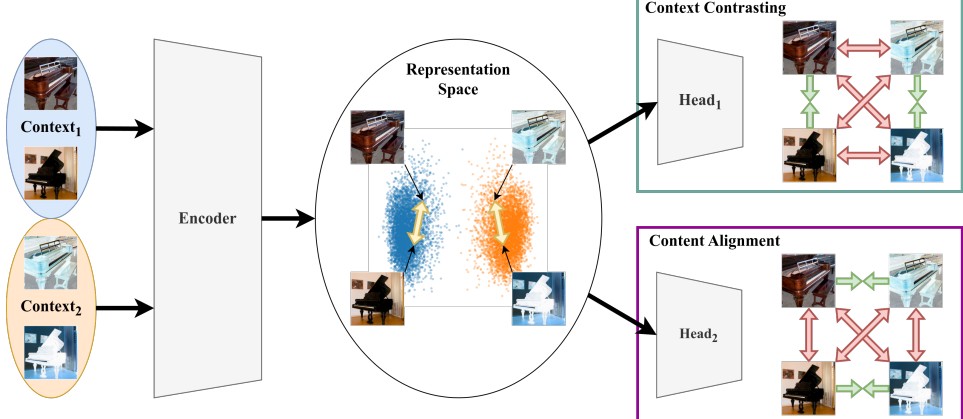

Figure 1: Overview of CON$_2$. Samples get context augmented and passed through an encoder. The *context contrasting* loss (Equation (2)) ensures context-specific representations (■ and ■ clusters) while the *content alignment* loss (Equation (3)) encourages a context independent structure (⟷) within each context cluster. We learn representations in a contrastive fashion, matching corresponding positive (⇥⇤) and discriminating between negative (⟷) pairs of representations separately for *context contrasting* and *content alignment*.

resentations within each cluster. Consequently, CON$_2$ ensures a highly informative structure within each cluster by preserving the relative normality of samples independent of their context.

Our main contributions include the definition of context augmentations to model invariances in normal data and the introduction of CON$_2$, which uses context augmentations to learn informative, tightly clustered representations of normal data. We further present the anomaly score function $\mathcal{S}_{\text{NND}}$ that measures the anomalousness of new samples given representations from CON$_2$. Additionally, we propose the $\mathcal{S}_{\text{LH}}$ anomaly score, which offers a more compute efficient alternative to our initial anomaly score. Finally, we demonstrate the advantage of modeling invariances of normal data in our experiments, where we present strong results when performing anomaly detection on specialized medical and more general natural image datasets.

In the next section, we provide an overview of related work and draw a comparison to our approach before proceeding to introduce our method.

## 2    RELATED WORK

Learning useful normal representations of high-dimensional data to perform anomaly detection has recently become a popular line of research. Prior work has tackled the problem from various angles, for instance, using hypersphere compression (Ruff et al., 2018). Other popular methods define pretext tasks such as learning reconstruction models (Chen et al., 2017; Zong et al., 2018; You et al., 2019) or predicting data transformations (Golan & El-Yaniv, 2018; Hendrycks et al., 2019b; Bergman & Hoshen, 2019). While these approaches had some success in the past, the learned representations are not very informative. On the other hand, methods learning more informative representations through self-supervised learning have recently been shown to improve over prior work (Sun et al., 2022; Sehwag et al., 2021).

Another line of work focuses on estimating the training density with the help of generative models, detecting anomalies as samples from low probability regions (An & Cho, 2015; Schlegl et al., 2019; Nachman & Shih, 2020; Mirzaei et al., 2022). However, these methods tend to generalize better to unseen distributions than to the observed training distribution (Nalisnick et al., 2018), which proves problematic for anomaly detection.

In addition to the traditional setting, where we assume training data without any labels, some recent works weaken this restriction and assume access to a limited number of labeled samples. This setting is called anomaly detection with Outlier Exposure (OE) (Hendrycks et al., 2019a), and it has

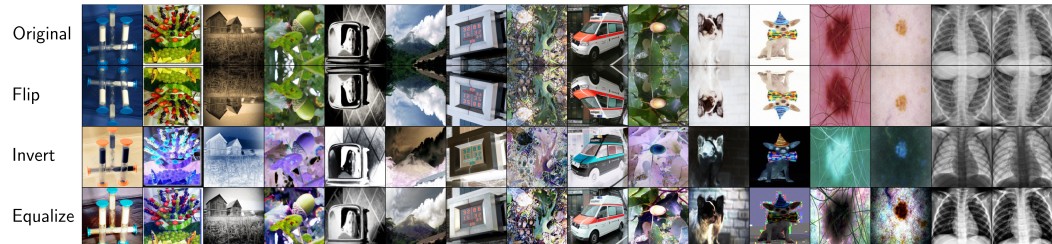

Figure 2: Examples of the context augmentations used throughout our experiments. *Flip* denotes vertical flipping, *Invert* denotes the transformation that replaces each pixel value $x$ with $1 - x$, and *Equalize* stands for histogram equalization. In our experiments, *Flip* and *Invert* fulfill alignment and distinctiveness for almost all datasets, while *Equalize* can sometimes violate distinctiveness.

been shown that already a few labeled samples can greatly boost performance over an unlabeled dataset (Ruff et al., 2020; Qiu et al., 2022; Liznerski et al., 2022). Using large, pretrained models as feature extractors is a special case of OE, as additional data is not explicitly accessible. Some approaches have been introduced that use representations from pretrained models directly in zero-shot fashion (Bergman et al., 2020; Liznerski et al., 2022; Jeong et al., 2023; Zhou et al., 2024), while others demonstrate the benefit of fine-tuning (Cohen & Avidan, 2022; Reiss & Hoshen, 2023; Li et al., 2023). OE has been very successful in the past, often outperforming traditional anomaly detection settings across many benchmarks, though at the cost of either requiring labeled samples or vast amounts of data for pretraining, which are both often not available or hard to obtain in more specialized domains.

Another setting that has recently gained popularity is out-of-distribution (OOD) detection. In OOD detection, we have additional information about our dataset in the form of labels. Anomaly detection is a special case of OOD detection with only a single label. While the problem is similar, most approaches that tackle OOD detection make specific use of a classifier trained on the dataset labels (Hendrycks & Gimpel, 2017; Lee et al., 2018; Wang et al., 2022), which cannot directly be applied in the anomaly detection setting, as training a classifier on a single class is not straightforward.

In comparison, our method operates in the traditional anomaly detection setting and can be applied to datasets without knowledge about anomalies. Further, while we assume access to a dataset containing normal samples, our method does not rely on additional labels, as they can be difficult and expensive to obtain, particularly in more specialized settings.

## 3 METHODS

In the following, we introduce the notion of context augmentations and then present our $\text{CON}_2$ objective, which leverages these augmentations to learn tightly clustered, informative representations. We then explain how to use these representations to detect anomalies at test time.

### 3.1 CONTEXT AUGMENTATION

The intuition behind context augmentation comes from the observation that certain transformations can augment a sample into another context, creating a distinct new view without altering its information content. For example, inverting an image, i.e., exchanging every pixel value $x$ with the value $1-x$, neither adds nor destroys any information (*alignment*) but instead allows us to observe the same sample from a different perspective. In the following, we want to learn these invariances while still being able to distinguish between the two transformations to learn symmetric representation clusters for both the original and the augmented sample space. In the previous example, this only works if the inverted version of an image does not naturally appear in the dataset already (*distinctiveness*). Otherwise, it is impossible to distinguish between original and transformed instances properly.

We define two requirements that let us determine whether a transformation is suitable as a context augmentation for a given dataset. Let $X \subset \mathcal{X}$ be our dataset, let $t_{\mathcal{C}} : \mathcal{X} \to \mathcal{X}$ be a data augmentation,

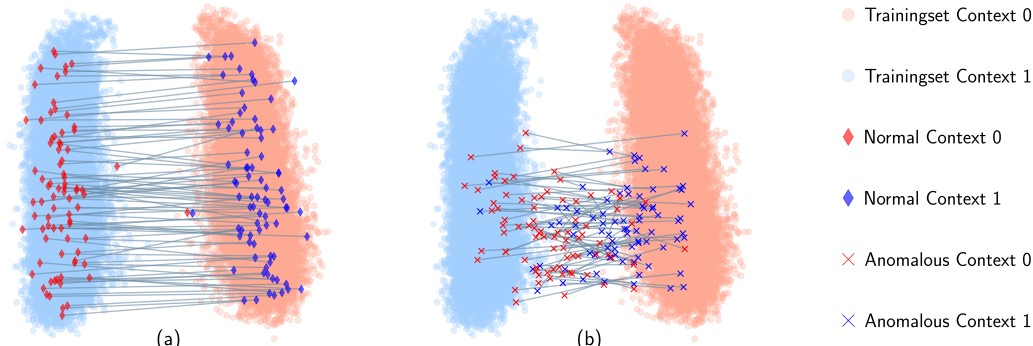

Figure 3: Two-dimensional PCA embedding of the train, normal test (a), and anomalous test samples (b) after training $\text{CON}_2$ on the *car* class of CIFAR10. The lines connecting representations mark embeddings corresponding to the same sample in different contexts. The parallel lines indicate that sample representations are positioned approximately at the same location across context clusters for the normal test samples, while anomalies do not exhibit the same invariances as normal samples and thus fail to adhere well to the learned structure.

and let $t_{\mathcal{C}}(X) = \{t_{\mathcal{C}}(\boldsymbol{x}) \mid \boldsymbol{x} \in X\}$ be the dataset transformed by $t_{\mathcal{C}}$. The function $t_{\mathcal{C}}$ is a *context augmentation* if it fulfills the following two properties:

**Distinctiveness** For two samples $\boldsymbol{x} \sim p_X$ from the original and $\boldsymbol{x}^{\mathcal{C}} \sim p_{t_{\mathcal{C}}(X)}$ from the context augmented data distribution, we have that $p_{t_{\mathcal{C}}(X)}(\boldsymbol{x}) \approx 0$ and $p_X(\boldsymbol{x}^{\mathcal{C}}) \approx 0$, i.e., there is a clear distinction between the original and the context augmented distribution after applying $t_{\mathcal{C}}$. For instance, if our normal class consists of images of melanoma, flipping the image violates distinctiveness, as melanoma can be photographed from any angle. Conversely, histogram equalizing or color inversion of the image satisfies distinctiveness, as the resulting color distribution is distinct from the original samples of such a dataset.

**Alignment** Let $\boldsymbol{x}, \boldsymbol{x}' \in X$, and let $\mathrm{d}(\boldsymbol{x}, \boldsymbol{x}')$ denote an appropriate similarity measure for samples in the input space. Then, we require that $\mathrm{d}(\boldsymbol{x}, \boldsymbol{x}') \approx \mathrm{d}(t_{\mathcal{C}}(\boldsymbol{x}), t_{\mathcal{C}}(\boldsymbol{x}'))$, i.e., originally similar normal samples should stay just as similar in the new context, meaning that the original and the context-augmented normal distributions should align. For instance, masking part of a torso x-ray image would violate alignment, as we could potentially remove important regions, such as the lungs, from the image altogether. On the other hand, two vertically flipped x-rays are as similar to each other as their original counterparts.

While it may be dataset-dependent whether a transformation, such as histogram equalization (*Equalize*), fulfills these conditions, there are transformations, such as vertical flipping (*Flip*) or color inversion (*Invert*), that seem to fulfill distinctiveness and alignment across a broad range of datasets. We present some examples of context augmentations in Figure 2.

## 3.2 CONTEXT CONTRASTING

Our method learns representations in a contrastive fashion (van den Oord et al., 2019). Contrastive learning is a popular approach for self-supervised representation learning. Typically, it relies on the definition of positive and negative pairs of samples and learns to maximize the similarity of representations of positive pairs while pushing apart representations of negative pairs. Popular contrastive approaches, such as SimCLR (Chen et al., 2020) or SupCon (Khosla et al., 2020), achieve this by incorporating a form of instance discrimination in their loss function. Here, we define the instance discrimination loss as

$$\ell(\boldsymbol{x}, \boldsymbol{x}', X) = -\log \frac{\exp\left(\mathrm{sim}(\boldsymbol{x}, \boldsymbol{x}')/\tau\right)}{\sum\limits_{\boldsymbol{x}'' \in X:\, \boldsymbol{x}'' \neq \boldsymbol{x}} \exp\left(\mathrm{sim}(\boldsymbol{x}, \boldsymbol{x}'')/\tau\right)}, \tag{1}$$

where we consider $\mathrm{sim}(\boldsymbol{x}, \boldsymbol{x}')$ to be the cosine similarity between two samples $\boldsymbol{x}, \boldsymbol{x}' \in X$. We refer to Appendix A.1 for more background about contrastive learning.

In the following, we present our novel $\text{CON}_2$ objective, which leverages the distinctiveness and alignment assumptions of context augmentations to learn informative representations, which we will later use for anomaly detection. Specifically, we apply distinctiveness to learn context-specific representation clusters. Alignment further allows us to distinguish samples from each other while encouraging a similar relative location of a sample across clusters. We present an example underlining this intuition in Figure 3, where we show our representation space after training a model with $\text{CON}_2$ on the samples of the *car* class of CIFAR10.

Assume a set of samples $X_{\text{train}}$, a context augmentation $t_\mathcal{C}$, a set of augmentations $\mathcal{T}$ that models invariances of the dataset like in Chen et al. (2020), and let

$$X_\mathcal{C} = \{(\boldsymbol{x}, 0) \mid \boldsymbol{x} \in X_{\text{train}}\} \cup \{(t_\mathcal{C}(\boldsymbol{x}), 1) \mid \boldsymbol{x} \in X_{\text{train}}\}$$

denote the dataset after applying context augmentation $t_\mathcal{C}$, labeling each sample with its context. For $t, t' \sim \mathcal{T}$ and $\boldsymbol{x}_i^\mathcal{C} \in X_\mathcal{C}$, let $\tilde{\boldsymbol{x}}_{2i} = t(\boldsymbol{x}_i^\mathcal{C})$ and $\tilde{\boldsymbol{x}}_{2i+1} = t'(\boldsymbol{x}_i^\mathcal{C})$ denote two transformations of the same context-augmented sample using random augmentations from $\mathcal{T}$ and let the set of all such pairs be

$$\tilde{X}_\mathcal{C} = \left\{ \left( t\left( \boldsymbol{x}^\mathcal{C} \right), y \right), \left( t'\left( \boldsymbol{x}^\mathcal{C} \right), y \right) \mid \left( \boldsymbol{x}^\mathcal{C}, y \right) \in X_\mathcal{C} \wedge t, t' \sim \mathcal{T} \right\}.$$

Further, we denote $f(\tilde{X}_\mathcal{C}) := \{(f(\boldsymbol{x}), y) \mid \boldsymbol{x} \in \tilde{X}_\mathcal{C}\}$ for any function $f$. $\text{CON}_2$ then consists of two parts, *context contrasting* and *content alignment*.

**Context Contrasting** By leveraging the distinctiveness property of context augmentations, we can learn tightly concentrated, context-specific representation clusters with our *context contrasting* loss. For a given sample $\boldsymbol{x}$, we derive its representation $g_\theta(\boldsymbol{x})$ using an encoder $g_\theta$. We then define the *context contrasting* loss as

$$\mathcal{L}_{\text{Context}}(\tilde{X}_\mathcal{C}) = \frac{1}{4N} \sum_{(\tilde{\boldsymbol{x}}_i, y_i) \in \tilde{X}_\mathcal{C}} \frac{1}{2N-1} \sum_{\substack{(\tilde{\boldsymbol{x}}_j, y_j) \in \tilde{X}_\mathcal{C} \\ \tilde{\boldsymbol{x}}_j \neq \tilde{\boldsymbol{x}}_i \wedge y_i = y_j}} \ell(f_\Phi(\tilde{\boldsymbol{x}}_i), f_\Phi(\tilde{\boldsymbol{x}}_j), f_\Phi(\tilde{X}_\mathcal{C})), \tag{2}$$

where $f_\Phi = h_\phi(g_\theta(\boldsymbol{x}))$ and $h_\phi$ is a projection head that gets discarded after training similar to Chen et al. (2020). Intuitively, context contrasting encourages representations of the same context to be clustered together while pushing other context clusters away, similar to the class representations in supervised contrastive learning (SupCon) (Khosla et al., 2020).

**Content Alignment** While $\mathcal{L}_{\text{Context}}$ allows us to learn context-dependent representation clusters, it does not enforce a specific structure within each cluster. To make the cluster structure more informative, $\text{CON}_2$ leverages the alignment property of context augmentations to align representations across clusters through context-independent instance discrimination. More specifically, let $\Lambda(i) = \{2i, 2i+1, 4i, 4i+1\}$, i.e., $\Lambda(i)$ corresponds to all indices[2] of samples in $\tilde{X}_\mathcal{C}$ which are augmentations of the original sample $\boldsymbol{x}_i \in X$. We then define the *content alignment* loss as

$$\mathcal{L}_{\text{Content}}(\tilde{X}_\mathcal{C}) = \frac{1}{N} \sum_{k=1}^{N} \frac{1}{12} \sum_{i \in \Lambda(k)} \sum_{j \in \Lambda(k) \smallsetminus i} \ell(f_\Psi(\tilde{\boldsymbol{x}}_i), f_\Psi(\tilde{\boldsymbol{x}}_j), f_\Psi(\tilde{X}_\mathcal{C})), \tag{3}$$

where $f_\Psi(\boldsymbol{x}) = h_\psi(g_\theta(\boldsymbol{x}))$, and $h_\psi$ denotes a projection head that is independent of $h_\phi$. Content alignment ensures that all representations of the same normal sample can be matched across different contexts, encouraging alignment of the representations within each context cluster.

Finally, we combine context contrasting and content alignment to our loss function $\text{CON}_2$, which enables us to learn *context-specific, content-aligned* representations of normality:

$$\mathcal{L}_{\text{Con}_2}(\tilde{X}_\mathcal{C}) = \mathcal{L}_{\text{Context}}(\tilde{X}_\mathcal{C}) + \alpha \mathcal{L}_{\text{Content}}(\tilde{X}_\mathcal{C}) \tag{4}$$

To account for the different scalings of $\mathcal{L}_{\text{Context}}$ and $\mathcal{L}_{\text{Content}}$, we introduce a weighting factor $\alpha \in \mathbb{R}^+$. Figure 1 provides a visual overview of how we apply $\text{CON}_2$ to learn representations.

---

[2] Indexing in correspondence to previous section. Strict ordering is not necessary.

## 3.3 ANOMALY DETECTION

In the anomaly detection setting, we typically assume an unlabeled training set containing predominantly normal samples, whereas we want to discriminate between normal and anomalous samples at test time (Ruff et al., 2021). To detect anomalies, we typically define an anomaly score function $\mathcal{S}$ that maps a given sample's representation onto a scalar, determining its anomalousness. We can then define a threshold on this anomaly score, predicting *anomaly* for samples above the threshold and *normal* for samples below. We provide additional background about the anomaly detection setting in Appendix A.2.

To detect anomalies using the representations of $\mathrm{CON_2}$, we define two anomaly score functions that measure how well a test sample adheres to the context representation clusters. One of the most popular and straightforward approaches to achieve this is a non-parametric nearest neighbor approach (Bergman et al., 2020; Sun et al., 2022). Our first score adopts a similar procedure using the cosine similarity. Specifically, let us define the cosine distance between the training set $X_{\text{train}}$ and a given test sample $\boldsymbol{x}$ with transformation $t$ as

$$s_{\mathrm{NND}}(\boldsymbol{x};t) = - \max_{\boldsymbol{x}' \in X_{\text{train}}} \frac{\langle g_\theta(t(\boldsymbol{x})), g_\theta(t(\boldsymbol{x}')) \rangle}{\|g_\theta(t(\boldsymbol{x}))\|\|g_\theta(t(\boldsymbol{x}'))\|}. \tag{5}$$

Intuitively, the better a new sample aligns with the context cluster given by augmentation $t$, the more likely we are to consider it to be normal. In turn, for samples with a lower cosine similarity, it seems to either be difficult to assign the correct context cluster, or they do not share much of the normal information within the correct context cluster. While this approach works well in practice, it is rather memory-inefficient, as we need to store the representations of all samples in $X_{\text{train}}$.

To adapt our approach to settings with resource constraints, we further introduce a likelihood-based score function $s_{\mathrm{LH}}$. To make this score function as lightweight as possible, we assume that representations within each context cluster are distributed according to a multivariate Gaussian, which allows us to efficiently estimate the empirical mean and covariance from the training set and evaluate the probability density to derive an anomaly score. Contrastive approaches typically tend to learn representations with relatively large norms, which may lead to numerical instabilities when estimating the covariance matrix. Our $s_{\mathrm{LH}}$ thus estimates the empirical mean and covariance on the normalized representations. In particular, let

$$Z_{train}^{(t)} = \left\{ \frac{g_\theta(t(\boldsymbol{x}))}{\|g_\theta(t(\boldsymbol{x}))\|} \mid \boldsymbol{x} \in X_{train} \right\} \tag{6}$$

be the normalized representations of the training set augmented with some augmentation $t$. We then compute the density of a multivariate normal distribution based on the empirical mean and covariance,

$$\overline{\mu}\left(Z_{train}^{(t)}\right) \text{ and } \overline{\Sigma}\left(Z_{train}^{(t)}\right). \tag{7}$$

We then define

$$s_{\mathrm{LH}}(\boldsymbol{x};t) = -\log\left(\mathcal{N}\left(\frac{g_\theta(t(\boldsymbol{x}))}{\|g_\theta(t(\boldsymbol{x}))\|} \mid \overline{\mu}\left(Z_{train}^{(t)}\right), \overline{\Sigma}\left(Z_{train}^{(t)}\right)\right)\right). \tag{8}$$

We further leverage that our model can differentiate between the two contexts and learns invariances across different augmentations from $\mathcal{T}$ by applying test-time augmentations, similar to previous works (Tack et al., 2020; Wang et al., 2023), which further improves our anomaly detection performance. More specifically, let $\mathcal{T}_{\text{test}} = \{t_1, \ldots, t_A\}$ be a set of $A$ test time augmentations. We define our final anomaly score functions $\mathcal{S}_{\{\mathrm{NND,LH}\}} : \mathcal{X} \to \mathbb{R}$ as

$$\mathcal{S}_{\{\mathrm{NND,LH}\}}(\boldsymbol{x}) = \frac{1}{A}\left(\sum_{i=1}^{A/2} s_{\{\mathrm{NND,LH}\}}(\boldsymbol{x};t_i) + \sum_{i=A/2}^{A} s_{\{\mathrm{NND,LH}\}}(\boldsymbol{x};t_i \circ t_{\mathcal{C}})\right), \tag{9}$$

where $\circ$ defines the composition of two functions (Peirce, 1852).

## 4 EXPERIMENTS

In the following, we present how $\mathrm{CON_2}$ allows us to learn highly informative representations of normality by incorporating prior knowledge about invariances of normal data. After briefly introducing our baselines, we demonstrate how we can leverage this knowledge in a realistic medical

setting, showcasing the applicability of our method to a specialized domain where prior knowledge about anomalies is typically hard to obtain. Further, we present the generality of our method by comparing it to baselines on popular natural image datasets in the one-class classification setting, where anomaly detection with $\text{CON}_2$ consistently exhibits strong performance across various settings. We refer to Appendices C and D for more details regarding the choice of hyperparameters and our datasets.

**Baselines**   We compare our work to various recent contrastive anomaly detection baselines, including SSD (Sehwag et al., 2021), CSI (Tack et al., 2020), and UniCon-HA (Wang et al., 2023). SSD works by learning representations using SimCLR and detecting anomalies with a Mahalanobis distance-based anomaly score. Similarly, CSI and UniCon-HA learn representations with SimCLR but additionally design synthetic anomalies using rotation transformations. CSI leverages these synthetic anomalies using an additional classifier to discriminate between normal and synthetic anomaly samples and detects anomalies at test time with a score that combines nearest neighbor distance, sample norm, and classifier confidence. UniCon-HA does not require an additional classifier but instead clusters all normal samples close to each other while minimizing the similarity of synthetic anomaly representations and normal training samples. UniCon-HA also modifies the instance discrimination loss to weight positive and negative pairs according to the distance between representations. It further introduces a hierarchical augmentation scheme that lets them apply their loss on different layers of their neural network architecture using layer-specific augmentation strategies. We also compare against a baseline that learns SimCLR embeddings and detects samples in nearest neighbor fashion similar to KNN+ (Sun et al., 2022), which was originally developed for out-of-distribution detection. Finally, we also compare to anomaly detection using CLIP (Radford et al., 2021; Liznerski et al., 2022), which detects anomalies by using a pretrained CLIP model and comparing image embeddings with text embeddings describing the normal class. Apart from CLIP-AD, we conduct all experiments with the ResNet18 architecture (He et al., 2016) to ensure comparability between methods.

## 4.1   MEDICAL ANOMALY DETECTION

In this experiment, we demonstrate how incorporating prior knowledge about invariances of normal data through context augmentations with $\text{CON}_2$ leads to strong anomaly detection performance on two challenging medical imaging datasets. We compare the performance of $\text{CON}_2$ with recent unsupervised anomaly detection methods. Additionally, we also compare to CLIP-AD (Liznerski et al., 2022), which relies on a pretrained CLIP model (Radford et al., 2021) and thus incorporates a form of outlier exposure as explained in Section 2.

Table 1: Anomaly detection results on two real-world medical imaging datasets. We train each model with three different seeds and report the mean ± standard deviation.

| Method | Score $\mathcal{S}$ | Pneumonia | Melanoma |
|---|---|---|---|
| CLIP-AD | $\mathcal{S}_{\text{CLIP}}$ | 71.2 | 77.2 |
| SimCLR | $\mathcal{S}_{\text{NND}}$ | $91.0_{\pm 0.9}$ | $72.9_{\pm 2.8}$ |
| SSD | $\mathcal{S}_{\text{Mahalanobis}}$ | $90.9_{\pm 0.2}$ | $79.0_{\pm 2.2}$ |
| CSI | $\mathcal{S}_{\text{CSI}}$ | $73.9_{\pm 1.6}$ | $92.3_{\pm 0.2}$ |
| UniCon-HA | $\mathcal{S}_{\text{UniCon}}$ | $86.4_{\pm 0.1}$ | $91.1_{\pm 0.8}$ |
| $\text{CON}_2$ (Equalize) | | $93.0_{\pm 0.3}$ | $94.0_{\pm 0.3}$ |
| $\text{CON}_2$ (Invert) | $\mathcal{S}_{\text{LH}}$ | $90.6_{\pm 1.0}$ | $93.0_{\pm 0.4}$ |
| $\text{CON}_2$ (Flip) | | $91.5_{\pm 0.6}$ | $92.9_{\pm 0.5}$ |
| $\text{CON}_2$ (Equalize) | | $\mathbf{93.9}_{\pm 0.3}$ | $\mathbf{94.5}_{\pm 0.2}$ |
| $\text{CON}_2$ (Invert) | $\mathcal{S}_{\text{NND}}$ | $91.1_{\pm 0.7}$ | $94.1_{\pm 0.4}$ |
| $\text{CON}_2$ (Flip) | | $92.8_{\pm 1.1}$ | $93.4_{\pm 1.1}$ |

We train $\text{CON}_2$ on the healthy samples of a real-world medical chest x-ray dataset (Kermany et al., 2018) and a melanoma imaging dataset (Javid, 2022), discriminating between unseen healthy and anomalous samples at test time. Here, we model invariances of normal samples with the three context augmentations *Flip*, *Invert*, and *Equalize* mentioned in Section 3.1. We run each experiment across three seeds, train on healthy samples, and apply our anomaly score functions to the representations of test samples to detect anomalies. We report the mean and standard deviation of the resulting area under the receiver operating characteristics curves (AUROC) in Table 1.

We can see that our $\text{CON}_2$ consistently exhibits a strong performance with both $\mathcal{S}_{\text{LH}}$ and $\mathcal{S}_{\text{NND}}$ across all three context augmentations. However, we note a significant performance decrease with *Flip* on

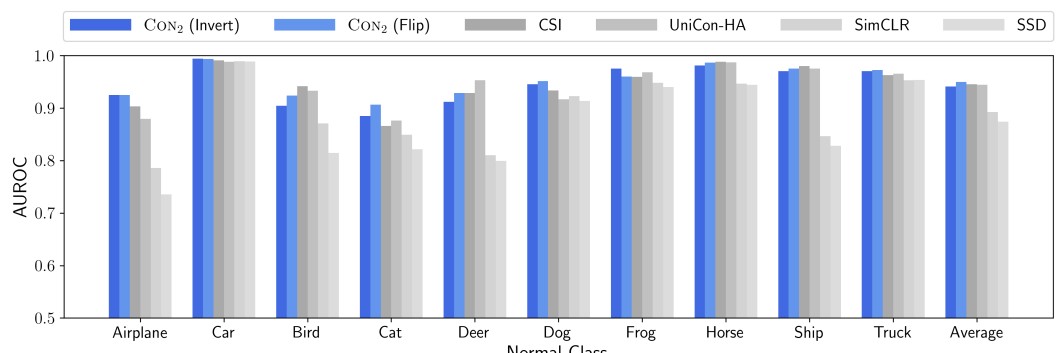

Figure 4: AUROCs of CIFAR10 when setting one class as normal and detecting the rest as anomalous. We compare $\text{CON}_2$ with the *Invert* and *Flip* context augmentations with $\mathcal{S}_{\text{NND}}$ to other contrastive anomaly detection methods. Both the *Invert* and *Flip* context augmentations fulfill our assumptions, resulting in good performances across all classes. Our method further outperforms our baselines in most classes. $\text{CON}_2$ with *Flip* has the highest average across all methods considered.

the Melanoma dataset. This performance decrease most likely stems from the fact that *Flip* violates distinctiveness on melanoma images as they could be taken from any angle. Apart from $\text{CON}_2$ (Flip) on Melanoma, our method outperforms all baselines, confirming that modeling invariances of normal data offers a clear advantage in specialized settings. We further note that the CLIP-AD method, which exhibits impressive performance on natural image datasets (see Appendix E.2), lacks behind most of our baselines, indicating that, even in the age of foundation models, unsupervised anomaly detection methods are still important in specialized domains.

## 4.2 NATURAL IMAGE BENCHMARKS

In addition to the results on the more specialized medical imaging domain, our method also exhibits robust performance on more traditional natural imaging benchmark datasets. In this experiment, we train $\text{CON}_2$ on the CIFAR10, CIFAR100 (Krizhevsky et al., 2009), ImageNet30 (Russakovsky et al., 2015; Hendrycks et al., 2019b), Dogs vs. Cats (Cukierski, 2013), and Muffin vs. Chihuahua (Cortinhas, 2023) datasets in the one-class classification setting (Ruff et al., 2021). In the one-class classification setting, we typically work on multi-class classification datasets where we consider one of the classes as the normal class and the rest as anomalies. In particular, we train our model on the training samples of the normal class and want to differentiate between unseen samples of this normal class and all other classes at test time. Here, we train each model across three seeds for each class of each dataset, reporting the mean and standard deviation of the resulting AUROCs.

On natural images, the *Equalize* context augmentation does not satisfy distinctiveness, as this transformation often results in scenes that seem slightly differently illuminated (see Figure 2 for some examples). We thus only present results of $\text{CON}_2$ with *Flip* and *Invert* context augmentations. In Section 4.1, we saw that the more efficient $\mathcal{S}_{\text{LH}}$ anomaly score exhibits relatively strong performance, however, $\mathcal{S}_{\text{NND}}$ typically performs slightly better and we thus only report $\mathcal{S}_{\text{NND}}$ in this section. Further, we note that CLIP-AD exhibits a strong performance on natural image datasets as, during training, CLIP has been exposed to samples that are similar to anomalies of this one-class classification setting. It is thus hard to compare CLIP-AD to our method and baselines, which were all trained without outlier exposure, on natural image datasets and we thus do not compare to CLIP-AD in this section. For completeness, we report the full results including including both scores, $\text{CON}_2$ with *Equalize*, and CLIP-AD results for all datasets in Appendix E.2.

In Figure 4, we compare the performance of $\text{CON}_2$ and our baselines across the different one-class settings of CIFAR10. $\text{CON}_2$ outperforms our baselines on almost all classes, where $\text{CON}_2$ (Flip) with an average AUROC of $95.3$ performs better than $\text{CON}_2$ (Invert), which exhibits an average AUROC of $94.6$. We suspect that *Invert* exhibits similar issues as *Equalize* in some instances, i.e., it may not always fully satisfy the distinctiveness assumption.

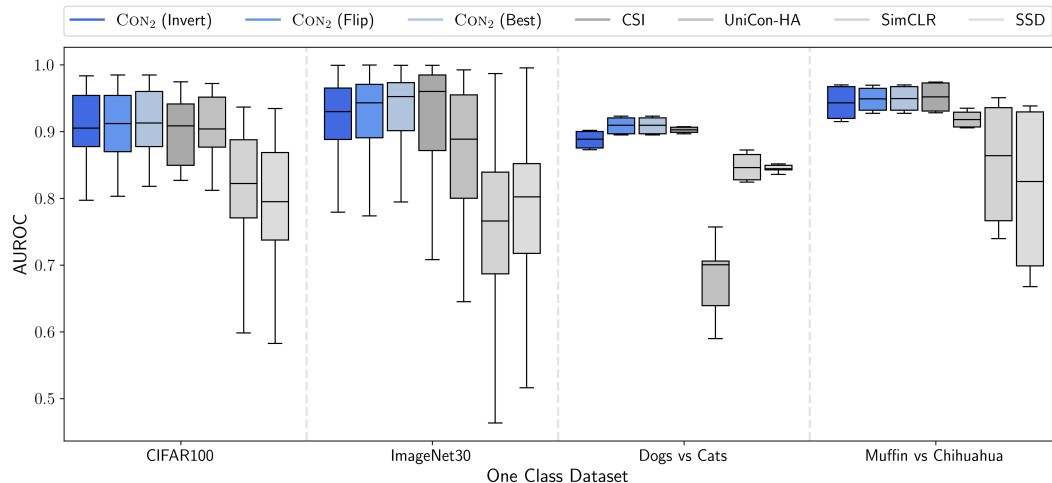

Figure 5: One class classification results for CIFAR100, ImageNet30, Dogs vs. Cats, and Muffin vs. Chihuahua. Our method consistently outperforms our baselines on CIFAR100 and Dogs vs. Cats while exhibiting more robust performance across different normal classes with a similar average performance to CSI on ImageNet30 and Muffin vs. Chihuahua. Additionally, we provide results including $CON_2$ (Best), which demonstrates how carefully selecting context augmentations satisfying the assumptions of Section 3.1 further improves the capabilities of anomaly detection with $CON_2$.

We further provide results on one-class CIFAR100, ImageNet30, Dogs vs. Cats, and Muffin vs. Chihuahua in Figure 5. There, in addition to the *Invert* and *Flip* context augmentation, we also provide results for $CON_2$ (Best), which selects the context augmentation individually for each class, depending on which satisfies alignment and distinctiveness better for the current normal class. Our method compares well against established baselines on natural images, matching or improving the state-of-the-art. Similar to what we saw on CIFAR10, $CON_2$ displays a robust performance across the board. Our approach outperforms baselines on CIFAR100 and Dogs vs. Cats while matching the performance on ImageNet30 and Muffin vs. Chihuahua while exhibiting much more consistent performance across different normal classes as can be seen from the much lower variance displayed in Figure 5. We can also see that selecting the context augmentation that best fits the normal class can improve the performance. However, we also achieve strong performance if the context augmentation violates alignment and distinctiveness on only some samples of the dataset. We provide further ablations, including experiments on applying multiple context augmentations, demonstrating that a single context augmentation is sufficient, and additional one-class classification results in Appendix E.

## 5 CONCLUSION

In this work, we presented a novel approach to anomaly detection, focusing on learning representations of normality by leveraging prior knowledge about invariances in the normal data rather than simulating anomalous data as in previous works. Employing knowledge about the invariances of normal data is more realistic and provides a stronger foundation for anomaly detection, particularly in specialized domains such as healthcare, where anomalous data is rare or hard to simulate accurately.

$CON_2$ learns dense, highly informative context clusters that capture the properties of normal data. These clusters provide rich representations and ensure that a sample's relative positioning is consistent across clusters, strengthening the model's ability to differentiate between normal and anomalous data. This results in a more structured representation space, making our approach well-suited for anomaly detection tasks with our anomaly score functions.

We demonstrated the efficacy of our approach on two real-world medical imaging datasets, where our method achieved impressive results. This highlights the applicability of $CON_2$ in safety-critical applications where robust anomaly detection is essential. Additionally, our approach exhibited strong performance on natural imaging datasets, consistently outperforming baseline methods,

demonstrating its versatility across different domains. Further, our method highlights the importance of domain-specific approaches in specialized fields like healthcare, where tailored models can outperform foundation model-based approaches such as CLIP-AD, despite their success in more general settings.

In conclusion, $CON_2$ represents a significant advancement in anomaly detection by learning structured representations of normal data without relying on anomalous data. This approach is particularly valuable in specialized, high-stakes settings, offering robust and effective solutions across various application domains.

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

# A  BACKGROUND

In this section, we provide some terminology for contrastive learning and background about the anomaly detection setting.

## A.1  CONTRASTIVE LEARNING

Recently, contrastive learning has emerged as a popular approach for representation learning (van den Oord et al., 2019; Chen et al., 2020). By design, contrastive learning has the capability to learn representations that are agnostic to certain invariances (von Kügelgen et al., 2021; Daunhawer et al., 2023), which makes contrastive learning a particularly interesting choice to learn informative representations of normal samples (Tack et al., 2020; Wang et al., 2023), as it allows us to incorporate prior knowledge about our data into the representing learning process in the form of data augmentations. More specifically, invariances are learned by forming positive and negative pairs over the training dataset by applying data augmentations that should retain the relevant content of a sample.

The goal of contrastive learning is to learn an encoding function $g_\theta(\boldsymbol{x})$, where representations of positive pairs of samples are close and negative pairs are far from each other. For a given pair of samples $\boldsymbol{x}, \boldsymbol{x}' \in X$, we can define the instance discrimination loss as (Sohn, 2016; Wu et al., 2018; van den Oord et al., 2019)

$$\ell(\boldsymbol{x}, \boldsymbol{x}', X) = -\log \frac{\exp\left(\mathrm{sim}(\boldsymbol{x}, \boldsymbol{x}')/\tau\right)}{\sum\limits_{\boldsymbol{x}'' \in X: \boldsymbol{x}'' \neq \boldsymbol{x}} \exp\left(\mathrm{sim}(\boldsymbol{x}, \boldsymbol{x}'')/\tau\right)} \, .$$

As mentioned in Section 3.2, we consider the function $\mathrm{sim}(\boldsymbol{x}, \boldsymbol{x}')$ to correspond to the cosine similarity between the two input vectors, as this is one of the most popular choices in the contrastive learning literature.

One of the most prominent contrastive methods is SimCLR (Chen et al., 2020), which creates positive pairs through sample augmentations. There exists a supervised extension called SupCon (Khosla et al., 2020), which incorporates class labels into the SimCLR loss. For a given set of augmentations $T$, a dataset $X = \{(\boldsymbol{x}_i, y_i)\}_{i=1}^{N}$, and an augmented dataset $\tilde{X}$ where $|\tilde{X}| = 2N$ and $(\tilde{\boldsymbol{x}}_{2i}, y_i)$, $(\tilde{\boldsymbol{x}}_{2i+1}, y_i) \in \tilde{X}$ denote two transformations of the same sample using random augmentations from $T$, SimCLR and SupCon introduce the following loss functions:

$$\mathcal{L}_{\mathrm{SimCLR}}(\tilde{X}) = \frac{1}{2N} \sum_{i=1}^{N} \left(\ell(f_\Theta(\tilde{\boldsymbol{x}}_{2i}), f_\Theta(\tilde{\boldsymbol{x}}_{2i+1}), f_\Theta(\tilde{X})) + \ell(f_\Theta(\tilde{\boldsymbol{x}}_{2i+1}), f_\Theta(\tilde{\boldsymbol{x}}_{2i}), f_\Theta(\tilde{X}))\right),$$

$$\mathcal{L}_{\mathrm{SupCon}}(\tilde{X}) = \sum_{(\tilde{\boldsymbol{x}}_i, y_i) \in \tilde{X}} \frac{1}{N(y_i) - 1} \sum_{\substack{(\tilde{\boldsymbol{x}}_j, y_j) \in \tilde{X}: \\ \tilde{\boldsymbol{x}}_j \neq \tilde{\boldsymbol{x}}_i \wedge y_i = y_j}} \ell(f_\Theta(\tilde{\boldsymbol{x}}_i), f_\Theta(\tilde{\boldsymbol{x}}_j), f_\Theta(\tilde{X})) \, .$$

Here, we denote $f_\Theta(\boldsymbol{x}) = h_{\theta'}(g_\theta(\boldsymbol{x}))$, where $\boldsymbol{z} = g_\theta(\boldsymbol{x})$ is a feature extractor and $h_{\theta'}(\boldsymbol{z})$ is a projection head that is typically only used during training (Chen et al., 2020). Further, we define $f_\Theta(\tilde{X}) = \{f_\Theta(\tilde{\boldsymbol{x}}) \mid (\tilde{\boldsymbol{x}}, y) \in \tilde{X}\}$ and $N(y) = |\{(\tilde{\boldsymbol{x}}_i, y_i) \mid (\tilde{\boldsymbol{x}}_i, y_i) \in \tilde{X} \wedge y_i = y\}|$ is the number of samples in $\tilde{X}$ with label $y$.

## A.2  ANOMALY DETECTION

In the anomaly detection setting, we are given an unlabeled dataset $\{\boldsymbol{x}_1, \ldots, \boldsymbol{x}_n\} = X \subset \mathcal{X}$, while assuming that most samples are normal, i.e., the dataset is practically free of outliers (Ruff et al., 2021). The goal is to learn a model from the given dataset that discriminates between normal and anomalous data at test time.

In this work, we assume the challenging case where our dataset is completely free of anomalies. Hence, we aim to discriminate between the normal class and a completely unobserved set of anomalies at test time. This setting is sometimes called one-class classification or novelty detection.

To achieve this goal, one straightforward approach is to approximate the distribution $p_{\mathcal{X}}(\boldsymbol{x})$ directly using generative models (An & Cho, 2015; Schlegl et al., 2019). Because we assume normal data to

Table 2: Average compute hours for the experiments for each dataset and method per run. SimCLR and SSD use the same representations, so we can evaluate both methods in one go and list their compute hours together.

| Method \ Dataset | CIFAR10 | CIFAR100 | ImageNet30 | Dogs vs. Cats | Muffin vs. Chihuahua | Pneumonia | Melanoma |
|---|---|---|---|---|---|---|---|
| SimCLR/SSD | 4 | 2 | 3 | 9 | 3 | 3 | 5 |
| CSI | 8 | 4 | 4 | 14 | 5 | 8 | 6 |
| UniCon-HA | 16 | 16 | 8 | 18 | 7 | 12 | 18 |
| $CON_2$ | 5 | 3 | 4 | 11 | 4 | 5 | 6 |

lie in high-density regions of $p_\mathcal{X}$, we can discriminate between normal and anomalous samples by applying a threshold function $p_\mathcal{X}(x) \leq \tau$, where $\tau \in \mathbb{R}$ is an often task-specific threshold (Bishop, 1994). As density-based approaches are often difficult to apply to high-dimensional data directly (Nalisnick et al., 2018), we follow a slightly different line of work.

In this paper, we focus on learning a function $g_\theta : \mathcal{X} \to \mathcal{Z}$ that provides us with representations that capture the normal attributes of samples in the dataset (Sehwag et al., 2021; Tack et al., 2020; Wang et al., 2023), by mapping normal samples close to each other in representation space. On the other hand, anomalies that lack the learned normal structure should be mapped to a different part of the representation space.

Given $g_\theta(x)$, a popular approach to detect anomalies is by defining a scoring function $\mathcal{S} : \mathcal{Z} \to \mathbb{R}$ (Breunig et al., 2000; Schölkopf et al., 2001; Tax & Duin, 2004; Liu et al., 2008). The score function maps a representation onto a metric that estimates the anomalousness of a sample. To identify anomalies at test time, we can use $\mathcal{S}$ similarly to the density $p_\mathcal{X}$, i.e., we consider a new sample $x$ to be normal if $\mathcal{S}(g_\theta(x)) \leq \tau$, whereas $\mathcal{S}(g_\theta(x)) > \tau$ means $x$ is an anomaly.

# B  COMPUTE & CODE

We run all our experiments on single GPUs on a compute cluster using a combination of RTX2080Ti, RTX3090, and RTX4090 GPUs. Each experiment can be run with 4 CPU workers and 16 GB of memory. We provide an overview of the compute for our experiments in Table 2. Our experiments are written using PyTorch (Ansel et al., 2024) with Lightning (Falcon & The PyTorch Lightning team, 2019).

In the following, we list for each of our methods and baselines how we arrive at results and which code we use.

**$CON_2$**: We implement $CON_2$ using PyTorch (Ansel et al., 2024) together with Lightning (Falcon & The PyTorch Lightning team, 2019). To evaluate our method, we use various open-source Python libraries such as NumPy (Harris et al., 2020), scikit-learn (Pedregosa et al., 2011), Pandas (McKinney, 2010; team, 2020), or SciPy (Virtanen et al., 2020). Parts of the implementation of the $CON_2$ objective are based on code provided by Khosla et al. (2020) (`https://github.com/HobbitLong/SupContrast`).

**SimCLR**: For this baseline, we implement SimCLR (Chen et al., 2020) and compute anomaly scores in a similar fashion as (Sun et al., 2022). For this baseline, we rely on similar packages as $CON_2$.

**SSD**: We use the same representations as for SimCLR but evaluate by following the procedure outlined in Sehwag et al. (2021).

**CSI**: To run experiments for CSI, we used the code provided in `https://github.com/alinlab/CSI`, implementing new dataloaders for the missing datasets.

**UniCon-HA**: We conducted experiments by running code provided by Wang et al. (2023) implementing new dataloaders for the missing datasets. We thank the authors for sharing their code with us.

## C  DATASETS

In the following, we provide details about preprocessing, sources, and licenses of the datasets we use in our experiments.

### PNEUMONIA

The Pneumonia dataset was originally published by Kermany et al. (2018) and consists of $5'863$ lung xrays, which are labeled with *Pneumonia* and *Normal* labels. We first resize images to 256 and apply center-cropping to feed 224×224 images to our model. We ran all our experiments on the Pneumonia dataset with a batch size of 128. The dataset can be downloaded from https://www.kaggle.com/datasets/paultimothymooney/chest-xray-pneumonia and is published under *CC BY 4.0* license.

### MELANOMA

We use the Melanoma dataset of Javid (2022), which consists of $10'600$ images of Melanoma labeled with being *benign* or *malignant*. We resize each image to size $128 \times 128$ before passing them to the model with batch size 128. The dataset is publicly available at https://www.kaggle.com/datasets/hasnainjaved/melanoma-skin-cancer-dataset-of-10000-images and is published under the *CC0: Public Domain* license.

### CIFAR10/CIFAR100

CIFAR10 and CIFAR100 are natural image datasets with $32 \times 32$ samples. Both datasets consist of a total of $60'000$ samples, with a total of 10 and 100 samples for CIFAR10 and CIFAR100, respectively. As CIFAR100 comes with only 600 samples per class, the dataset authors additionally define a set of 20 superclasses, aggregating 5 labels each. In our one-class classification experiments on CIFAR100 we use the superclasses to ensure a manageable number of runs and a sufficient amount of training data. We ran all our experiments on CIFAR10 and CIFAR100 with a batch size of 512. Both datasets were published by Krizhevsky et al. (2009) and can be downloaded from https://www.cs.toronto.edu/~kriz/cifar.html. To the best of our knowledge, these datasets come without a license.

### IMAGENET30

The ImageNet30 dataset is a subset of the original ImageNet dataset (Russakovsky et al., 2015). It was created by Hendrycks et al. (2019b) for the purpose of one-class classification. The dataset consists of $42'000$ natural images where each is labeled with one of 30 classes. We preprocess the dataset by resizing the shorter edge to 256 pixels, from which we randomly crop a $224 \times 224$ image patch every time we load an image for training. We ran all our experiments on ImageNet with a batch size of 128. The dataset can be downloaded from https://github.com/hendrycks/ss-ood, which comes with the MIT License. Further, while we could not find a license for ImageNet, terms of use are provided on https://image-net.org/.

### DOGS VS. CATS

The Dogs vs. Cats was originally introduced in a Kaggle challenge by Microsoft Research (Cukierski, 2013) and consists of $25'000$ images of cats and dogs. We preprocess the dataset by resizing the shorter edge to 128 pixels and then perform center cropping, feeding the resulting $128 \times 128$ image to our model. We ran all our experiments on Dogs vs. Cats with a batch size of 256. The dataset can be downloaded from https://www.kaggle.com/competitions/dogs-vs-cats/data. To the best of our knowledge, there is no official license for the dataset, but the Kaggle page points to the Kaggle Competition rules https://www.kaggle.com/competitions/dogs-vs-cats/rules in the license section.

CHIHUAHUA VS. MUFFIN

The Chihuahua vs. Muffin dataset consists of $6'000$ images scraped from Google Images. We preprocess the dataset similar to ImageNet30, resizing the shorter edge of the images to 128 pixels while feeding random $128 \times 128$ sized image crops to the model during training. We ran all our experiments on Chihuahua vs. Muffin with a batch size of 256. The dataset was published by Cortinhas (2023) and can be downloaded from `https://www.kaggle.com/datasets/samuelcortinhas/muffin-vs-chihuahua-image-classification/data`. According to the datasets Kaggle page, the dataset is licensed under *CC0: Public Domain*.

In addition to the preprocessing mentioned above, we normalize each image with a mean and standard deviation of $0.5$ after applying the augmentations of $\text{CON}_2$.

## D  EXPERIMENTAL DETAILS

**Setting**   We evaluate our method in the so-called one-class classification setting (Ruff et al., 2021). More specifically, during training we assume to have access to only the normal (healthy) class. At test time, the goal is to detect whether a new sample stems from the normal class seen during training or whether it seems anomalous, i.e., deviates from the training distribution.

**Metrics**   Typically, there is a high-class imbalance between normal and anomalous samples in the one-class classification setting. Further, setting an appropriate threshold for the anomaly score is often task-dependent. Therefore, a popular approach to evaluating the performance of anomaly detection methods is to use the area under the receiver operator characteristic curve (AUROC) (Ruff et al., 2021). This metric is threshold agnostic and robust to class imbalance.

For all our experiments, we report mean and standard deviation over three seeds per class of the dataset. Note that the average results of a dataset are aggregated over different one-class classification settings, one per class of the dataset.

**Hyperparameters**   Similar to our method, all baselines make use of test-time augmentations. By default, both CSI and UniCon-HA use 40 test time augmentations, which we adopt for all baselines. In our experiments, we set the augmentation class $\mathcal{T}$ to the set of augmentations introduced by Chen et al. (2020). For the context augmentation, we experiment with vertical flips (Flip), inverting the pixels of an image (Invert), i.e., $t_{\text{Invert}}(\boldsymbol{x}_{ij}) = 1 - \boldsymbol{x}_{ij}$, and histogram equalization (Equalize), see Figure 2 for an illustration.

We choose hyperparameters for $\text{CON}_2$ based on their performance on the CIFAR10 dataset and keep them constant across all experiments. We linearly anneal the hyperparameter $\alpha$ in $\mathcal{L}_{\text{CON}_2}$ from 0 to 1 over the course of training to encourage the model to first learn the context-specific cluster structure while gradually aligning representations over the course of training. We optimize our loss using the AdamW optimizer (Loshchilov & Hutter, 2019) with $\beta_1 = 0.9$, $\beta_2 = 0.999$, weight decay $\lambda = 0.001$, and using a learning rate of $10^{-3}$ with a cosine annealing (Loshchilov & Hutter, 2017) schedule. We run all experiments for 2048 epochs.

## E  ABLATIONS

In this section, we provide some additional experiments going beyond only two context clusters (Appendix E.1) and a more detailed overview of the results on natural images (Section 4.2).

### E.1  MULTIPLE CONTEXT AUGMENTATIONS

Our formulation in Section 3.1 can easily be extended beyond only one additional context by slightly adjusting $\mathcal{L}_{\text{Context}}$. However, in addition to a loss in efficiency due to requiring more memory, we did not find additional context augmentations to provide a performance benefit, as can be seen in Figure 6. There, we ran an ablation with different numbers of context augmentations on different classes of CIFAR10 and ImageNet30. In particular, we trained the adapted $\text{CON}_2$ loss for 2, 3, 4, 5, 6, 7, and 8 context augmentations, which we derived by combining *Flip*, *Invert*, and *Equalize* from our previous experiments. Adding more augmentations does not seem to harm cases where we

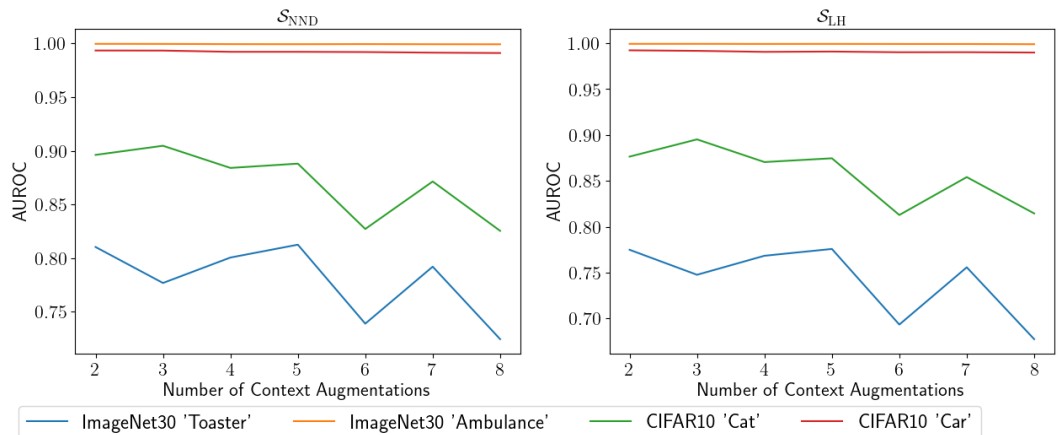

Figure 6: Ablation illustrating the effect of adding more context augmentations. While the performance of well-performing normal classes, such as ImageNet30 *Ambulance* or CIFAR10 *Car* stays consistent when adding more augmentations, we see a decrease for normal classes such as ImageNet30 *Toaster* or CIFAR10 *Cat* that already perform poor to begin with.

Table 3: One class classification results for CIFAR10, CIFAR100, ImageNet30, Dogs vs. Cats, and Muffin vs. Chihuahua. For each dataset, we train models over three different seeds per dataset class. We report mean and standard deviation over all the different one-class settings per dataset.

| Method | Score | CIFAR10 | CIFAR100 | ImageNet30 | Dogs vs. Cats | Muffin vs. Chihuahua |
|---|---|---|---|---|---|---|
| CLIP-AD (OE) | $\mathcal{S}_{\text{CLIP}}$ | $98.5_{\pm1.0}$ | $95.1_{\pm2.7}$ | $99.9_{\pm0.2}$ | $99.7_{\pm0.2}$ | $98.6_{\pm2.0}$ |
| SimCLR | $\mathcal{S}_{\text{NND}}$ | $89.2_{\pm6.7}$ | $81.6_{\pm8.5}$ | $74.7_{\pm12.2}$ | $84.7_{\pm2.2}$ | $85.2_{\pm9.8}$ |
| SSD | $\mathcal{S}_{\text{Mahalanobis}}$ | $87.4_{\pm8.1}$ | $79.2_{\pm9.4}$ | $76.8_{\pm13.0}$ | $84.5_{\pm0.6}$ | $81.3_{\pm13.1}$ |
| CSI | $\mathcal{S}_{\text{CSI}}$ | $94.6_{\pm4.0}$ | $90.2_{\pm4.9}$ | $92.3_{\pm8.1}$ | $90.3_{\pm0.4}$ | $95.2_{\pm2.3}$ |
| UniCon-HA | $\mathcal{S}_{\text{UniCon-HA}}$ | $94.4_{\pm4.0}$ | $90.9_{\pm4.4}$ | $85.5_{\pm12.0}$ | $67.9_{\pm6.2}$ | $91.9_{\pm1.3}$ |
| $\text{CON}_2$ (Equalize) | $\mathcal{S}_{\text{LH}}$ | $91.1_{\pm5.8}$ | $86.4_{\pm6.0}$ | $85.1_{\pm13.0}$ | $79.5_{\pm1.7}$ | $85.8_{\pm11.2}$ |
|  | $\mathcal{S}_{\text{NND}}$ | $91.5_{\pm5.6}$ | $87.8_{\pm4.8}$ | $86.2_{\pm12.1}$ | $83.2_{\pm1.2}$ | $88.3_{\pm8.3}$ |
| $\text{CON}_2$ (Invert) | $\mathcal{S}_{\text{LH}}$ | $93.7_{\pm4.3}$ | $89.7_{\pm5.2}$ | $90.2_{\pm9.4}$ | $87.9_{\pm0.6}$ | $91.7_{\pm4.3}$ |
|  | $\mathcal{S}_{\text{NND}}$ | $94.6_{\pm3.6}$ | $90.9_{\pm4.7}$ | $90.7_{\pm8.9}$ | $88.8_{\pm1.4}$ | $94.3_{\pm2.7}$ |
| $\text{CON}_2$ (Flip) | $\mathcal{S}_{\text{LH}}$ | $94.7_{\pm3.5}$ | $89.8_{\pm5.2}$ | $89.4_{\pm11.1}$ | $90.3_{\pm0.8}$ | $93.4_{\pm2.6}$ |
|  | $\mathcal{S}_{\text{NND}}$ | $95.3_{\pm2.9}$ | $90.8_{\pm4.8}$ | $90.5_{\pm10.4}$ | $90.9_{\pm1.4}$ | $94.9_{\pm2.0}$ |

experience good performance in the first place, however, we observe a diminishing performance for slightly more challenging classes.

### E.2 ADDITIONAL ONE CLASS CLASSIFICATION RESULTS

In Section 4.2, we present figures with results of $\text{CON}_2$ on CIFAR10, CIFAR100, ImageNet30, Cats vs. Dogs, and Muffin vs. Chihuahua. For completeness, we also present a table containing the full results of $\text{CON}_2$ on all three context augmentations mentioned in Section 3.1 and both scores from Section 3.3 in Table 3. We further present results aggregated over individual one-class classification settings of CIFAR10, CIFAR100, ImageNet30, Cats vs. Dogs, and Muffin vs. Chihuahua for $\text{CON}_2$ on all datasets. We present results for CIFAR10 in Table 4, for all 20 superclasses of CIFAR100 in Table 5, for ImageNet30 in Table 6, for Dogs vs. Cats in Table 7, and for Muffin vs. Chihuahua in Table 8.

### E.3 CONTRIBUTIONS OF INDIVIDUAL LOSS PARTS

We perform an ablation study where we evaluate the contribution of the loss components $\mathcal{L}_{\text{Content}}(\cdot)$ (see also Equation (3)) and $\mathcal{L}_{\text{Context}}(\cdot)$ (see also Equation (2)). We see that combining the content loss with the novel context loss leads to performance improvements on both evaluated datasets, Pneumonia and Melanoma.

Table 4: AUROCS for each class of CIFAR10 for both of our scores when applying the Flip context augmentation. For each setting, we evaluated our method across three seeds.

| Normal Class | $CON_2$ (Equalize) | | $CON_2$ (Invert) | | $CON_2$ (Flip) | |
|---|---|---|---|---|---|---|
| | $\mathcal{S}_{LH}$ | $\mathcal{S}_{NND}$ | $\mathcal{S}_{LH}$ | $\mathcal{S}_{NND}$ | $\mathcal{S}_{LH}$ | $\mathcal{S}_{NND}$ |
| 0 | $89.7_{\pm0.4}$ | $91.3_{\pm0.6}$ | $90.3_{\pm0.5}$ | $\mathbf{92.5}_{\pm0.3}$ | $90.3_{\pm0.9}$ | $\mathbf{92.5}_{\pm0.7}$ |
| 1 | $98.4_{\pm0.0}$ | $98.5_{\pm0.2}$ | $99.2_{\pm0.1}$ | $\mathbf{99.4}_{\pm0.0}$ | $99.3_{\pm0.0}$ | $\mathbf{99.4}_{\pm0.0}$ |
| 2 | $87.0_{\pm0.2}$ | $88.7_{\pm0.5}$ | $88.3_{\pm0.4}$ | $90.5_{\pm0.1}$ | $91.6_{\pm0.6}$ | $\mathbf{92.4}_{\pm0.6}$ |
| 3 | $77.7_{\pm1.0}$ | $78.5_{\pm2.1}$ | $86.2_{\pm0.5}$ | $88.5_{\pm0.4}$ | $88.7_{\pm0.3}$ | $\mathbf{90.7}_{\pm0.3}$ |
| 4 | $90.2_{\pm0.8}$ | $90.1_{\pm0.6}$ | $90.8_{\pm0.3}$ | $91.2_{\pm0.5}$ | $\mathbf{93.1}_{\pm0.5}$ | $92.9_{\pm1.1}$ |
| 5 | $88.5_{\pm0.3}$ | $87.7_{\pm0.6}$ | $93.9_{\pm0.2}$ | $94.5_{\pm0.3}$ | $94.7_{\pm0.2}$ | $\mathbf{95.1}_{\pm0.1}$ |
| 6 | $96.4_{\pm0.3}$ | $96.0_{\pm0.1}$ | $97.3_{\pm0.2}$ | $\mathbf{97.5}_{\pm0.2}$ | $96.5_{\pm0.1}$ | $96.0_{\pm0.3}$ |
| 7 | $95.6_{\pm0.3}$ | $95.9_{\pm0.3}$ | $97.6_{\pm0.1}$ | $98.1_{\pm0.1}$ | $98.6_{\pm0.0}$ | $\mathbf{98.7}_{\pm0.0}$ |
| 8 | $93.2_{\pm0.3}$ | $94.4_{\pm0.2}$ | $96.3_{\pm0.1}$ | $97.1_{\pm0.1}$ | $97.0_{\pm0.3}$ | $\mathbf{97.5}_{\pm0.2}$ |
| 9 | $94.1_{\pm0.3}$ | $93.5_{\pm0.1}$ | $96.7_{\pm0.3}$ | $97.0_{\pm0.3}$ | $97.1_{\pm0.1}$ | $\mathbf{97.3}_{\pm0.1}$ |

Table 5: AUROCS for each superclass of CIFAR100 for both of our scores when applying the Flip context augmentation. For each setting, we evaluated our method across three seeds.

| Normal Class | $CON_2$ (Equalize) | | $CON_2$ (Invert) | | $CON_2$ (Flip) | |
|---|---|---|---|---|---|---|
| | $\mathcal{S}_{LH}$ | $\mathcal{S}_{NND}$ | $\mathcal{S}_{LH}$ | $\mathcal{S}_{NND}$ | $\mathcal{S}_{LH}$ | $\mathcal{S}_{NND}$ |
| 0 | $85.6_{\pm1.4}$ | $\mathbf{86.8}_{\pm0.6}$ | $84.9_{\pm0.5}$ | $86.2_{\pm0.4}$ | $84.5_{\pm0.6}$ | $86.0_{\pm0.7}$ |
| 1 | $84.9_{\pm1.2}$ | $87.2_{\pm1.7}$ | $87.4_{\pm0.5}$ | $\mathbf{88.1}_{\pm0.3}$ | $87.3_{\pm0.7}$ | $87.9_{\pm0.9}$ |
| 2 | $94.4_{\pm0.5}$ | $94.8_{\pm0.6}$ | $95.8_{\pm0.2}$ | $\mathbf{96.4}_{\pm0.1}$ | $94.8_{\pm0.3}$ | $95.0_{\pm0.4}$ |
| 3 | $80.6_{\pm1.4}$ | $83.9_{\pm0.8}$ | $88.4_{\pm0.2}$ | $89.2_{\pm0.7}$ | $90.3_{\pm0.7}$ | $\mathbf{90.4}_{\pm1.0}$ |
| 4 | $95.4_{\pm0.8}$ | $95.9_{\pm0.9}$ | $96.6_{\pm0.1}$ | $\mathbf{97.2}_{\pm0.1}$ | $95.0_{\pm0.1}$ | $96.1_{\pm0.2}$ |
| 5 | $70.7_{\pm2.4}$ | $78.5_{\pm3.1}$ | $82.8_{\pm0.3}$ | $\mathbf{86.2}_{\pm0.7}$ | $81.8_{\pm1.3}$ | $85.9_{\pm1.0}$ |
| 6 | $80.3_{\pm1.2}$ | $80.5_{\pm0.9}$ | $89.6_{\pm0.7}$ | $\mathbf{90.4}_{\pm0.5}$ | $90.3_{\pm0.8}$ | $\mathbf{90.4}_{\pm1.3}$ |
| 7 | $88.6_{\pm0.8}$ | $89.5_{\pm0.5}$ | $88.5_{\pm0.1}$ | $\mathbf{89.7}_{\pm0.2}$ | $86.5_{\pm0.8}$ | $87.6_{\pm0.6}$ |
| 8 | $88.9_{\pm0.2}$ | $89.7_{\pm0.4}$ | $91.3_{\pm0.4}$ | $\mathbf{92.3}_{\pm0.2}$ | $90.8_{\pm0.4}$ | $91.6_{\pm0.4}$ |
| 9 | $90.2_{\pm1.4}$ | $91.5_{\pm1.5}$ | $94.5_{\pm0.4}$ | $\mathbf{95.7}_{\pm0.4}$ | $94.8_{\pm0.2}$ | $95.6_{\pm0.3}$ |
| 10 | $82.5_{\pm4.2}$ | $84.0_{\pm4.5}$ | $88.9_{\pm0.3}$ | $\mathbf{90.4}_{\pm0.5}$ | $85.4_{\pm0.9}$ | $88.5_{\pm0.7}$ |
| 11 | $87.3_{\pm1.6}$ | $87.6_{\pm1.5}$ | $90.3_{\pm0.2}$ | $90.9_{\pm0.3}$ | $\mathbf{91.1}_{\pm0.3}$ | $\mathbf{91.1}_{\pm0.5}$ |
| 12 | $86.8_{\pm1.0}$ | $87.8_{\pm1.7}$ | $88.7_{\pm0.7}$ | $89.7_{\pm0.3}$ | $91.0_{\pm0.2}$ | $\mathbf{91.5}_{\pm0.2}$ |
| 13 | $82.7_{\pm1.7}$ | $\mathbf{85.5}_{\pm0.7}$ | $80.7_{\pm1.1}$ | $84.3_{\pm1.2}$ | $82.6_{\pm0.6}$ | $84.4_{\pm1.1}$ |
| 14 | $90.9_{\pm0.8}$ | $90.3_{\pm0.6}$ | $95.7_{\pm0.3}$ | $96.2_{\pm0.2}$ | $96.7_{\pm0.2}$ | $\mathbf{97.2}_{\pm0.1}$ |
| 15 | $81.1_{\pm1.0}$ | $\mathbf{82.0}_{\pm0.2}$ | $79.8_{\pm0.4}$ | $80.2_{\pm0.5}$ | $80.5_{\pm0.3}$ | $81.2_{\pm0.8}$ |
| 16 | $83.8_{\pm0.6}$ | $85.2_{\pm0.4}$ | $85.3_{\pm0.5}$ | $\mathbf{87.2}_{\pm0.5}$ | $85.6_{\pm0.6}$ | $86.1_{\pm0.8}$ |
| 17 | $95.3_{\pm1.5}$ | $95.8_{\pm1.5}$ | $98.0_{\pm0.1}$ | $\mathbf{98.3}_{\pm0.1}$ | $97.7_{\pm0.4}$ | $\mathbf{98.3}_{\pm0.3}$ |
| 18 | $91.1_{\pm1.5}$ | $90.2_{\pm2.1}$ | $94.9_{\pm0.2}$ | $95.4_{\pm0.2}$ | $95.9_{\pm0.0}$ | $\mathbf{96.1}_{\pm0.1}$ |
| 19 | $86.7_{\pm0.3}$ | $88.2_{\pm0.6}$ | $92.5_{\pm0.3}$ | $93.9_{\pm0.2}$ | $93.8_{\pm0.3}$ | $\mathbf{94.7}_{\pm0.3}$ |

Table 6: AUROCS for each class of ImageNet30 for both of our scores when applying the Flip context augmentation. For each setting, we evaluated our method across three seeds.

| Normal Class | $CON_2$ (Equalize) | | $CON_2$ (Invert) | | $CON_2$ (Flip) | |
|---|---|---|---|---|---|---|
| | $\mathcal{S}_{LH}$ | $\mathcal{S}_{NND}$ | $\mathcal{S}_{LH}$ | $\mathcal{S}_{NND}$ | $\mathcal{S}_{LH}$ | $\mathcal{S}_{NND}$ |
| 0 | $91.0_{\pm0.5}$ | $92.7_{\pm0.5}$ | $\mathbf{94.8}_{\pm0.7}$ | $94.7_{\pm0.9}$ | $92.1_{\pm0.7}$ | $92.4_{\pm1.3}$ |
| 1 | $97.8_{\pm0.2}$ | $98.7_{\pm0.2}$ | $98.5_{\pm0.2}$ | $99.2_{\pm0.1}$ | $99.1_{\pm0.1}$ | $\mathbf{99.5}_{\pm0.1}$ |
| 2 | $99.6_{\pm0.1}$ | $99.5_{\pm0.1}$ | $\mathbf{99.9}_{\pm0.0}$ | $\mathbf{99.9}_{\pm0.0}$ | $\mathbf{99.9}_{\pm0.0}$ | $\mathbf{99.9}_{\pm0.0}$ |
| 3 | $82.8_{\pm1.1}$ | $82.1_{\pm2.0}$ | $\mathbf{82.9}_{\pm1.2}$ | $79.1_{\pm0.9}$ | $82.6_{\pm0.6}$ | $82.6_{\pm1.0}$ |
| 4 | $90.5_{\pm0.3}$ | $90.1_{\pm0.7}$ | $95.0_{\pm0.1}$ | $94.8_{\pm0.2}$ | $94.7_{\pm0.2}$ | $\mathbf{95.6}_{\pm0.5}$ |
| 5 | $91.0_{\pm2.2}$ | $93.3_{\pm1.7}$ | $93.6_{\pm0.3}$ | $94.4_{\pm0.2}$ | $94.8_{\pm0.4}$ | $\mathbf{96.7}_{\pm0.4}$ |
| 6 | $94.8_{\pm0.7}$ | $95.5_{\pm0.3}$ | $97.1_{\pm0.1}$ | $\mathbf{98.0}_{\pm0.1}$ | $96.4_{\pm0.2}$ | $96.9_{\pm0.1}$ |
| 7 | $67.9_{\pm1.7}$ | $68.7_{\pm0.3}$ | $77.1_{\pm1.0}$ | $\mathbf{78.5}_{\pm0.8}$ | $75.8_{\pm1.5}$ | $76.5_{\pm1.6}$ |
| 8 | $95.4_{\pm0.2}$ | $95.7_{\pm0.4}$ | $93.4_{\pm0.7}$ | $92.7_{\pm1.3}$ | $96.6_{\pm0.5}$ | $\mathbf{96.9}_{\pm0.3}$ |
| 9 | $74.8_{\pm1.0}$ | $77.7_{\pm1.1}$ | $86.8_{\pm0.3}$ | $\mathbf{88.9}_{\pm0.2}$ | $84.9_{\pm0.8}$ | $86.6_{\pm0.6}$ |
| 10 | $97.8_{\pm0.2}$ | $97.8_{\pm0.1}$ | $99.2_{\pm0.1}$ | $\mathbf{99.3}_{\pm0.0}$ | $99.0_{\pm0.2}$ | $99.0_{\pm0.2}$ |
| 11 | $82.4_{\pm1.4}$ | $82.4_{\pm1.6}$ | $85.4_{\pm0.3}$ | $84.4_{\pm0.5}$ | $89.2_{\pm1.0}$ | $\mathbf{90.2}_{\pm0.7}$ |
| 12 | $90.3_{\pm0.4}$ | $93.0_{\pm0.6}$ | $95.5_{\pm0.2}$ | $97.3_{\pm0.1}$ | $96.6_{\pm0.2}$ | $\mathbf{97.6}_{\pm0.3}$ |
| 13 | $91.9_{\pm0.7}$ | $91.8_{\pm0.5}$ | $95.2_{\pm0.3}$ | $\mathbf{95.5}_{\pm0.2}$ | $94.0_{\pm0.5}$ | $94.0_{\pm0.3}$ |
| 14 | $85.1_{\pm0.1}$ | $86.6_{\pm0.7}$ | $91.6_{\pm0.6}$ | $91.9_{\pm0.5}$ | $93.3_{\pm0.3}$ | $\mathbf{94.2}_{\pm0.2}$ |
| 15 | $90.9_{\pm0.9}$ | $89.3_{\pm1.3}$ | $\mathbf{93.9}_{\pm0.6}$ | $92.7_{\pm0.2}$ | $93.2_{\pm1.9}$ | $93.1_{\pm2.0}$ |
| 16 | $96.6_{\pm0.4}$ | $97.5_{\pm0.3}$ | $98.9_{\pm0.1}$ | $99.2_{\pm0.1}$ | $99.0_{\pm0.1}$ | $\mathbf{99.5}_{\pm0.2}$ |
| 17 | $45.7_{\pm1.1}$ | $51.2_{\pm2.3}$ | $59.0_{\pm1.0}$ | $\mathbf{62.9}_{\pm1.0}$ | $50.9_{\pm1.1}$ | $55.2_{\pm0.6}$ |
| 18 | $78.4_{\pm0.7}$ | $80.3_{\pm0.9}$ | $89.2_{\pm0.3}$ | $89.8_{\pm0.5}$ | $92.2_{\pm0.6}$ | $\mathbf{93.1}_{\pm0.4}$ |
| 19 | $59.6_{\pm2.4}$ | $61.4_{\pm2.7}$ | $75.1_{\pm0.7}$ | $\mathbf{76.3}_{\pm0.5}$ | $67.1_{\pm3.4}$ | $68.5_{\pm3.7}$ |
| 20 | $86.3_{\pm1.0}$ | $86.2_{\pm0.6}$ | $92.2_{\pm0.5}$ | $93.0_{\pm0.6}$ | $94.2_{\pm0.4}$ | $\mathbf{94.9}_{\pm0.6}$ |
| 21 | $86.5_{\pm0.3}$ | $87.0_{\pm0.9}$ | $95.7_{\pm0.3}$ | $\mathbf{96.4}_{\pm0.1}$ | $95.7_{\pm0.2}$ | $96.2_{\pm0.2}$ |
| 22 | $95.3_{\pm0.5}$ | $94.4_{\pm0.8}$ | $96.7_{\pm0.5}$ | $96.1_{\pm0.3}$ | $97.3_{\pm0.3}$ | $\mathbf{97.4}_{\pm0.2}$ |
| 23 | $94.1_{\pm0.4}$ | $94.5_{\pm0.4}$ | $96.3_{\pm0.2}$ | $96.7_{\pm0.3}$ | $96.4_{\pm0.1}$ | $\mathbf{96.9}_{\pm0.2}$ |
| 24 | $72.0_{\pm1.3}$ | $73.7_{\pm0.7}$ | $90.3_{\pm0.3}$ | $\mathbf{92.4}_{\pm0.5}$ | $88.3_{\pm1.0}$ | $90.9_{\pm1.1}$ |
| 25 | $83.3_{\pm2.2}$ | $\mathbf{85.3}_{\pm1.8}$ | $84.6_{\pm1.3}$ | $84.1_{\pm1.5}$ | $73.6_{\pm2.3}$ | $74.1_{\pm1.0}$ |
| 26 | $95.1_{\pm0.2}$ | $\mathbf{95.2}_{\pm0.5}$ | $93.3_{\pm0.4}$ | $92.3_{\pm0.6}$ | $89.1_{\pm0.8}$ | $88.7_{\pm0.9}$ |
| 27 | $91.6_{\pm0.9}$ | $91.3_{\pm1.0}$ | $96.3_{\pm0.2}$ | $96.8_{\pm0.3}$ | $97.0_{\pm0.3}$ | $\mathbf{97.5}_{\pm0.2}$ |
| 28 | $57.3_{\pm1.5}$ | $61.4_{\pm2.5}$ | $69.0_{\pm1.3}$ | $72.1_{\pm1.8}$ | $73.8_{\pm2.8}$ | $\mathbf{77.7}_{\pm3.4}$ |
| 29 | $87.0_{\pm1.1}$ | $\mathbf{91.4}_{\pm1.1}$ | $88.5_{\pm1.7}$ | $90.8_{\pm2.1}$ | $86.1_{\pm1.0}$ | $91.3_{\pm1.4}$ |

Table 7: AUROCS for the two classes "Dog" and "Cat" for both of our scores when applying the Flip context augmentation. For each setting, we evaluated our method across three seeds.

| Normal Class | $CON_2$ (Equalize) | | $CON_2$ (Invert) | | $CON_2$ (Flip) | |
|---|---|---|---|---|---|---|
| | $\mathcal{S}_{LH}$ | $\mathcal{S}_{NND}$ | $\mathcal{S}_{LH}$ | $\mathcal{S}_{NND}$ | $\mathcal{S}_{LH}$ | $\mathcal{S}_{NND}$ |
| 0 | $78.4_{\pm1.7}$ | $84.1_{\pm0.9}$ | $88.3_{\pm0.1}$ | $90.0_{\pm0.2}$ | $91.0_{\pm0.1}$ | $\mathbf{92.1}_{\pm0.2}$ |
| 1 | $80.6_{\pm0.9}$ | $82.3_{\pm0.5}$ | $87.4_{\pm0.4}$ | $87.6_{\pm0.4}$ | $\mathbf{89.7}_{\pm0.4}$ | $\mathbf{89.7}_{\pm0.2}$ |

Table 8: AUROCS for the two classes "Muffin" and "Chihuahua" for both of our scores when applying the Flip context augmentation. For each setting, we evaluated our method across three seeds.

| Normal Class | $CON_2$ (Equalize) | | $CON_2$ (Invert) | | $CON_2$ (Flip) | |
|---|---|---|---|---|---|---|
| | $\mathcal{S}_{LH}$ | $\mathcal{S}_{NND}$ | $\mathcal{S}_{LH}$ | $\mathcal{S}_{NND}$ | $\mathcal{S}_{LH}$ | $\mathcal{S}_{NND}$ |
| 0 | $94.0_{\pm0.8}$ | $94.4_{\pm1.1}$ | $95.6_{\pm0.3}$ | $\mathbf{96.8}_{\pm0.2}$ | $95.8_{\pm0.2}$ | $96.7_{\pm0.3}$ |
| 1 | $73.6_{\pm1.3}$ | $79.2_{\pm0.3}$ | $87.8_{\pm0.5}$ | $91.8_{\pm0.3}$ | $91.1_{\pm0.7}$ | $\mathbf{93.1}_{\pm0.3}$ |

| Method | Score | Pneumonia | Melanoma |
|---|---|---|---|
| $\mathcal{L}_{\text{Content}}$ | | $92.3 \pm 0.9$ | $92.8 \pm 0.2$ |
| $\mathcal{L}_{\text{Context}}$ | $\mathcal{S}_{\text{LH}}$ | $79.9 \pm 1.3$ | $92.7 \pm 0.5$ |
| $\mathcal{L}_{\text{Con}_2}$ | | $\mathbf{93.0} \pm 0.3$ | $\mathbf{94.0} \pm 0.3$ |
| $\mathcal{L}_{\text{Content}}$ | | $89.6 \pm 0.4$ | $93.1 \pm 0.3$ |
| $\mathcal{L}_{\text{Context}}$ | $\mathcal{S}_{\text{NND}}$ | $81.4 \pm 1.6$ | $92.5 \pm 0.8$ |
| $\mathcal{L}_{\text{Con}_2}$ | | $\mathbf{93.9} \pm 0.3$ | $\mathbf{94.5} \pm 0.2$ |

Table 9: Evaluating the individual loss terms against each other when using the Equalize context augmentation on the medical datasets.

