# OpenReview forum: "Anomaly Detection by Context Contrasting"
_ICLR.cc/2025/Conference — Submitted to ICLR 2025_

### Official Review · Reviewer_yEuR · 2024-10-17

**Soundness:** 2
**Presentation:** 3
**Contribution:** 2
**Rating:** 5
**Confidence:** 3

**Summary:**

This paper proposes an anomaly detection method that uses context contrastive learning to learn useful representations of normal data.
Specifically, this paper proposes context augmentations that are transformations satisfying two properties (distinctiveness and alignment).
Using them, the invariance of normal patterns can be encoded well into the representations. The experiments with image datasets show the effectiveness of the proposed approach.

**Strengths:**

- The paper is generally well-written and is easy to follow.
- The proposed method is simple and thus may be easy for practitioners to use.
- The experiments with medical image data show that the proposed approach outperforms the existing contrastive anomaly detection methods.

**Weaknesses:**

- Since there are existing contrastive anomaly detection methods as described in the experiments, I think the proposed approach's novelty is not outstanding. Also, the differences between context and other ordinary transformations are not clearly explained. For example, can we use context transformations in the same way as ordinary transformations? (The same applies in reverse). I want to know the key differences between context augmentations and ordinary transformations.
- There are many anomaly detection approaches other than the contrasting-based approach described in Section 2. Thus, this paper would be improved by showing the effectiveness of the proposed method through experimental comparisons with these methods.
- In Figure 4, the significant difference between the proposed and existing methods is unclear.

**Questions:**

- The problem settings only have unlabelled or normal data. How are the hyperparameters of the proposed method practically determined?
- Can the proposed method be applied to domains other than the image domain?

---

> ### Author Response · Authors · 2024-11-19
>
> Dear reviewer yEuR,
>
> We would like to thank you again for your detailed review. Below, we answer your questions and address your concerns.
>
> > Since there are existing contrastive anomaly detection methods as described in the experiments, I think the proposed approach's novelty is not outstanding. Also, the differences between context and other ordinary transformations are not clearly explained. For example, can we use context transformations in the same way as ordinary transformations? (The same applies in reverse). I want to know the key differences between context augmentations and ordinary transformations.
>
>
> The proposed Con2 contains two parts. In the first part, the samples are projected into different contexts using context augmentations. In the second part, the samples are structured according to their content given a context.
> For contexts, augmentations should fulfill the two properties, *distinctiveness* and *alignment*. For content augmentations, any augmentation typically used for instance-discrimination can be leveraged.
> The main difference between context and content augmentations lies in their role within our loss function. Content augmentations are designed to capture sample invariances, while context augmentations define a new representation space where all normal samples remain consistent with one another (*alignment*) but appear distinct from their original forms (*distinctiveness*).
>
> ---
>
> > There are many anomaly detection approaches other than the contrasting-based approach described in Section 2. Thus, this paper would be improved by showing the effectiveness of the proposed method through experimental comparisons with these methods.
>
>
> Contrastive learning-based approaches are indeed only one of the multiple approaches to anomaly detection. However, contrastive learning approaches have continuously outperformed other self-supervised learning approaches, e.g., [1, 2], when applied to the contamination-free AD setting. Our proposed method itself is also a contrastive learning method. These are the two reasons we only compare it to other contrastive learning-based anomaly detection methods.
>
> ---
>
> > In Figure 4, the significant difference between the proposed and existing methods is unclear.
>
> We clearly outperform even the best baseline approaches for 6 out of 10 different normal classes (airplane, car, cat, dog, frog, truck) with at least one of the proposed context augmentations. The flip augmentation also outperforms all baseline methods when comparing their average performance across all classes. Additionally, we highlight Con2's clear advantages on specialized medical datasets, where the approach of generating synthetic anomalous samples, like in CSI and UniCON, is more challenging.
>
> ---
>
> > The problem settings only have unlabelled or normal data. How are the hyperparameters of the proposed method practically determined?
>
>
> We follow best practices in this work and choose the hyperparameters based on the validation set, which can contain only normal or a combination of normal samples and anomalies. For our algorithm, we chose the default parameters from previous works, e.g., the augmentation strengths of the SimCLR augmentations for the content objective. We only anneal the alpha value that weights the different loss functions because of the different scaling between the content and the context loss (see more details in response to Reviewer h61g).
>
>
> ---
>
> > Can the proposed method be applied to domains other than the image domain?
>
> Yes, it can. The proposed Con2 objective is general and can be applied broadly across multiple modalities. However, we would need to define an objective for the content loss either by leveraging natural invariances in the data, e.g., time-series data, or designing meaningful augmentations. In addition, we also need to define the context object with augmentations that reflect *distinctiveness* and *alignment*. We view this as a promising direction for future work and plan to explore it in follow-up studies.
>
> ---
>
> Thank you again for your feedback. We welcome any additional suggestions, questions, or requests for information and encourage further discussions.

---

> > ### Comment · Reviewer_yEuR · 2024-11-25
> >
> > Thank you for your response.
> >
> > >Contrastive learning-based approaches are indeed only one of the multiple approaches to anomaly detection. However, contrastive learning approaches have continuously outperformed other self-supervised learning approaches, e.g., [1, 2], when applied to the contamination-free AD setting. Our proposed method itself is also a contrastive learning method. These are the two reasons we only compare it to other contrastive learning-based anomaly detection methods.
> >
> > In addition to contrastive learning and self-supervised learning, many other methods such as density ratio estimation-based, autoencoders-based, and SVDD-based methods have been proposed for anomaly detection as described in Sec. 2. Since this paper focuses on anomaly detection, I think it is necessary to experimentally compare with these other approaches.
> >
> > >We follow best practices in this work and choose the hyperparameters based on the validation set, which can contain only normal or a combination of normal samples and anomalies. For our algorithm, we chose the default parameters from previous works, e.g., the augmentation strengths of the SimCLR augmentations for the content objective. We only anneal the alpha value that weights the different loss functions because of the different scaling between the content and the context loss (see more details in response to Reviewer h61g).
> >
> > It was stated that 'We choose hyperparameters for CON2 based on their performance on the CIFAR10 dataset ' in Line 952. Does this mean that labeled CIFAR10 data was used? If yes, since the proposed method is designed for settings where anomalous data cannot be used, I think the hyperparameters should also be chosen without using anomalous data.

---

> > > ### Author Response · Authors · 2024-11-26
> > >
> > > We appreciate your comments and are happy to address your concerns. Below, we provide detailed responses to each point.
> > >
> > >
> > > **Comparison to non-contrastive methods**
> > >
> > > We appreciate the reviewer’s emphasis on exploring a diverse set of anomaly detection (AD) methods. However, we believe it is unnecessary to include additional comparisons simply for the sake of expanding the results table. As noted in several works [4, 10, 11], SVDD-based approaches, even when incorporating outlier exposure [12], consistently underperform in comparison to contrastive SSL-based AD methods. Similarly, density-ratio estimation and reconstruction-based approaches, while effective in simpler AD scenarios, have been shown to struggle in complex and high-dimensional settings [11, 12]. For these reasons, it has become standard practice in the literature to omit comparisons to such approaches when presenting state-of-the-art contrastive learning-based AD methods.
> > >
> > > We hope the reviewer agrees that our decision to focus on meaningful comparisons enhances the clarity and rigor of our evaluation.
> > >
> > > ---
> > >
> > > **Hyperparameter selection**
> > >
> > > We agree with the reviewer that hyperparameters should be chosen without reliance on anomalous data. In our rebuttal, we have outlined how the loss on a validation set of normal samples can be used for this purpose. For transparency, we clarify in our manuscript that while we initially tuned hyperparameters on the CIFAR10 dataset (a publicly accessible dataset), these parameters were not adjusted for any of the other datasets. Thus, all reported results on datasets other than CIFAR10 were obtained without access to their anomalies. We demonstrate that the chosen set of hyperparameters allows robust performance over a diverse range of datasets and serves as a good starting point to further tune the hyperparameters on a new dataset.
> > >
> > > We emphasize that our method is capable of tuning hyperparameters solely based on the loss on a validation set of normal samples independent of anomalous data. This aligns with the principles of contamination-free anomaly detection and can be employed in practical use cases.
> > >
> > > ---
> > >
> > > We hope that our clarifications address your concerns and further demonstrate the robustness and practicality of our approach. Thank you again for your constructive feedback, and we look forward to any additional questions or suggestions you may have.
> > >
> > >
> > > [10] Bergman, Liron, and Yedid Hoshen. ‘Classification-Based Anomaly Detection for General Data’. International Conference on Learning Representations, 2020, https://openreview.net/forum?id=H1lK_lBtvS.
> > >
> > > [11] Hendrycks, Dan, et al. "Using self-supervised learning can improve model robustness and uncertainty." Advances in neural information processing systems 32 (2019).
> > >
> > > [12] Ruff, Lukas, et al. ‘Deep Semi-Supervised Anomaly Detection’. International Conference on Learning Representations, 2020, https://openreview.net/forum?id=HkgH0TEYwH.
> > >
> > > [13] Nalisnick, Eric, et al. ‘Do Deep Generative Models Know What They Don’t Know?’ International Conference on Learning Representations, 2019, https://openreview.net/forum?id=H1xwNhCcYm.
> > >
> > > [14] Havtorn, Jakob D., et al. "Hierarchical vaes know what they don’t know." International Conference on Machine Learning. PMLR, 2021.

---

### Official Review · Reviewer_h61g · 2024-10-28

**Soundness:** 2
**Presentation:** 2
**Contribution:** 2
**Rating:** 3
**Confidence:** 4

**Summary:**

This paper proposes an anomaly detection method, where encoders are trained such that the images with different context have separated representations while keeping the relationship among images within each context. With experiments using medical and natural images, the effectiveness of the proposed method is evaluated.

**Strengths:**

This paper proposes a new anomaly detection method with context augmentations.

The experiments with medical images.

**Weaknesses:**

The novelty of the proposed method is limited.

The advantage of the proposed method is unclear compared with the existing anomaly detection methods with data augmentation by flipping, inverting, and equalizing.

**Questions:**

Preparing contexts might require prior knowledge for anomaly. Can any context that fullfills distinctiveness and alignment be useful in the proposed method?

How can we tune alpha?

How to chose LH or NND?

What are the advantages of the proposed method compared with the existing anomaly detection methods with data augmentation?

The proposed method depends on the context transformation to be used (Equalize, Invert, Flip). How can we choose appropriate transformation for given applications?

What does happen when all transformations are included in the proposed method?

What happens when alpha is fixed at zero or one?

Why the context-specific cluster structure is important for anomaly detection? In the experiments, alpha is linearly annealed from zero to one.

---

> ### Author Response · Authors · 2024-11-19
>
> Dear reviewer h61g,
>
> We would like to thank you again for your detailed review. Below, we answer your questions and address your concerns.
>
>
> > The novelty of the proposed method is limited.
>
> We respectfully disagree with the reviewer’s opinion on this point. We introduce a novel contrastive learning objective that can leverage different contexts, defined by *distinctiveness* and *alignment*, to improve anomaly detection performance. Unlike previous works, which try to simulate anomalous samples, the proposed work is based on modeling properties of the normal samples, which is much simpler.
>
> ---
>
> > The advantage of the proposed method is unclear compared with the existing anomaly detection methods with data augmentation by flipping, inverting, and equalizing.
> > What are the advantages of the proposed method compared with the existing anomaly detection methods with data augmentation?
>
> The primary advantage of our method is its independence from prior knowledge about anomalies, unlike previous approaches that rely on simulating anomalous samples [1,2]. Instead, we leverage prior knowledge about the normal data available in the training set, making our method particularly suitable for specialized domains such as medicine.
>
> Further, our assumptions of *distinctiveness* and *alignment* of context-augmented samples are directly incorporated into the two loss terms of Con2. These assumptions enable us to learn compact and informative representation spaces while avoiding challenges like hypersphere collapse encountered in previous methods [4].
>
> ---
>
> > Preparing contexts might require prior knowledge for the anomaly. Can any context that fullfills distinctiveness and alignment be useful in the proposed method?
>
> The design of context augmentations only requires knowledge about the normal class and not anomalies. State-of-the-art works like UniCon and CSI try to model anomalies, which relies on having additional knowledge about anomalous data. The proposed method instead creates context augmentations based on knowledge about normal samples, which we have access to during training and, hence, is much simpler. In principle, any augmentation that satisfies the two properties can be utilized with Con2. However, it is important to note that no single augmentation typically fulfills both assumptions for every sample in a dataset. As a result, we cannot guarantee that all feasible augmentations will perform identically.
>
> ---
>
> > How can we tune alpha?
>
>
> We did not experience the need for tuning the hyperparameter $\alpha$ extensively. We only anneal $\alpha$ from 0 to 1 because of the different ranges between context and content loss. This difference arises because the context loss incorporates multiple positive samples, which results in different lower bounds for the two loss terms. Without annealing, the content loss would dominate the Con2 loss, causing the context loss to be neglected for a significant portion of the training. This imbalance can result in suboptimal local minima. However, in our opinion, there is no need to further tune this hyperparameter as the described setting works well across all our datasets.
>
> ---
>
> > How to chose LH or NND?
>
>
> The NND score is prominent among representation learning-based approaches, e.g., [1, 2, 5, 6]. It works well (see the reported results, incl. the updated results for the Melanoma dataset). However, computing the NND for the full set can be computationally expensive, especially for larger training datasets. Therefore, we propose a simple and lightweight likelihood-based score that works almost as well and is suitable for resource-constrained settings.
>
> ---
>
>
> > The proposed method depends on the context transformation to be used (Equalize, Invert, Flip). How can we choose appropriate transformation for given applications?
>
>
> The new method, Con2, highlights that augmentations – if they fulfill the properties of *distinctiveness* and *alignment* – can be used to map samples into different context spaces. Equalize, Flip, and Invert are example augmentations that we found to work well in the image domain, both on natural and medical images. However, there might be additional augmentation classes that work equally well. Different augmentations might be needed for other modalities, and we will continue exploring additional modalities in future work. Designing context augmentations is about finding transformations such that the transformed and original sample follow the properties of *distinctiveness* and *alignment* (see also section 3.1 in the original manuscript).
> We are not sure what you mean by “for given applications”? We would appreciate a clarification of the question.

---

> > ### Author Response · Authors · 2024-11-19
> >
> > > What does happen when all transformations are included in the proposed method?
> >
> > We also combined multiple context augmentations to increase the number of different contexts during the development of the method. However, we found that the method performs best with only two contexts, and adding contexts does not lead to additional performance gains. We also reported the corresponding ablation study in the appendix, section E.1 (Multiple Context Augmentations), of the original manuscript.
> >
> > ---
> >
> > > What happens when alpha is fixed at zero or one?
> > > Why the context-specific cluster structure is important for anomaly detection? In the experiments, alpha is linearly annealed from zero to one.
> >
> >
> > Thank you for these two questions. We performed an additional ablation study, analyzing the anomaly detection performance with only one of the two losses. In the following table, we can see the results for the individual loss terms:
> >
> > | Method                   | Score                | Pneumonia         | Melanoma          |
> > |--------------------------|----------------------|--------------------|--------------------|
> > | $\mathcal{L}_{\text{Content}}$ | $\mathcal{S}_{\text{LH}}$ | $92.3 \pm 0.9$    | $92.8 \pm 0.2$    |
> > | $\mathcal{L}_{\text{Context}}$ | $\mathcal{S}_{\text{LH}}$ | $79.9 \pm 1.3$    | $92.7 \pm 0.5$    |
> > | $\mathcal{L}_{\text{Con}_2}$   | $\mathcal{S}_{\text{LH}}$ | **$93.0 \pm 0.3$** | **$94.0 \pm 0.3$** |
> > | $\mathcal{L}_{\text{Content}}$ | $\mathcal{S}_{\text{NND}}$ | $89.6 \pm 0.4$    | $93.1 \pm 0.3$    |
> > | $\mathcal{L}_{\text{Context}}$ |  $\mathcal{S}_{\text{NND}}$ | $81.4 \pm 1.6$    | $92.5 \pm 0.8$    |
> > | $\mathcal{L}_{\text{Con}_2}$   | $\mathcal{S}_{\text{NND}}$ | **$93.9 \pm 0.3$** | **$94.5 \pm 0.2$** |
> >
> >
> > We see that having the combination of context and content loss performs the best across the two datasets and anomaly score functions. It is interesting to see that the context loss consistently improves the performance compared to the content-only objective.
> >
> > We also added the additional results to Appendix E.3.
> > The context-specific cluster structure serves as a convenient way to learn compact, informative representations, which are generally advantageous for robust AD [7]. This approach also avoids the issue of hypersphere collapse encountered in prior methods [4].
> >
> > We anneal $\alpha$ from 0 to 1 because of the different scaling of the two losses. Annealing further improves the stability of the training.
> >
> > ---
> >
> > Thank you again for your feedback. We welcome any additional suggestions, questions, or requests for information and encourage further discussions.

---

> > > ### Author Response · Authors · 2024-11-28
> > > **Reminder**
> > >
> > > Dear Reviewer h61g,
> > >
> > > We appreciate your detailed feedback. We have carefully addressed your concerns and believe our responses further demonstrate the value and impact of our work. If there are any additional points you’d like us to clarify, we would be happy to provide further details.

---

### Official Review · Reviewer_ENmc · 2024-10-31

**Soundness:** 3
**Presentation:** 2
**Contribution:** 2
**Rating:** 5
**Confidence:** 4

**Summary:**

This paper aims to address anomaly detection in a setting where no outliers or pre-trained models are available. The author aims to use a set of augmentations such that augmented images are (i) distinct - images from different augmentations are separated (ii) aligned - augmentations are distance preserving. They use a contrastive loss to ensure (ii). Anomalies are scored using this representation in one of two ways: kNN and Mahalanobis distance. Results are better than other SSL methods in standard datasets and better than CLIP-AD on two medical datasets.

**Strengths:**

1. Detailing the desiderata of augmentation in terms of distinctiveness and alignment is interesting, and to the best of my knowledge, novel.
2. Results are strong w.r.t. to other SSL methods.
3. Figure 3 is a very nice illustration of the main technical point.

**Weaknesses:**

1. There is somewhat misleading about covering pre-trained based methods as "outlier detection". There is a vast number of papers showing that pre-trained features, while reliant on seeing images not in the training set, are much more generalizable than outlier exposure. Namely, strong visual features go beyond purely adding prior knowledge about specific anomalies [1][2].
2. Only one none-SSL method is compared, and not the most standard one
3. The main results table, justifying the focus on SSL, shows only two datasets.

Minor comments:
4. The paper claims (line 46) "without assuming prior knowledge about anomalies". I find that misleading, prior are always needed in order to generalize anything to an unseen test. Even more so in unsupervised settings. In this case, the priors are the inductive biases induced by the augmentations among other things.
5. I found Figure.1 more confusing than helpful.

[1] Reiss, Tal, et al. "Panda: Adapting pretrained features for anomaly detection and segmentation." Proceedings of the IEEE/CVF Conference on Computer Vision and Pattern Recognition. 2021.
[2] Roth, Karsten, et al. "Towards total recall in industrial anomaly detection." Proceedings of the IEEE/CVF conference on computer vision and pattern recognition. 2022.

**Questions:**

1. Please compare to more pre-trained-based methods. E.g., (Cohen & Avidan, 2022; Reiss & Hoshen, 2023;
Li et al., 2023) cited by your paper.
2. Please extend this comparison to more medical datasets. E.g., ChestX-ray14, HAM10000.

---

> ### Author Response · Authors · 2024-11-19
>
> Dear reviewer ENmc,
>
> We would like to thank you again for your detailed review. Below, we answer your questions and address your concerns.
>
> > There is somewhat misleading about covering pre-trained based methods as "outlier detection". There is a vast number of papers showing that pre-trained features, while reliant on seeing images not in the training set, are much more generalizable than outlier exposure. Namely, strong visual features go beyond purely adding prior knowledge about specific anomalies [1][2].
> > Only one none-SSL method is compared, and not the most standard one
>
> We focus on SSL methods due to their superior performance in AD, when we only have access to normal samples. Our focus is on leveraging AD for the medical domain, where – in general – we do not have access to large, internet-scale datasets that would allow the pretraining of methods. Exploring pre-trained models in more specialized areas is an interesting direction for future work.
>
> ---
>
> > The main results table, justifying the focus on SSL, shows only two datasets.
> Table 1 shows the results of the experiments in the medical domain, which we see as the most targeted application for the proposed Con2 objective. However, we present additional convincing results on general benchmark datasets such as CIFAR 10, CIFAR 100, Imagenet 30, Cats vs. Dogs, and Muffin vs. Chihuahua.
>
> > The paper claims (line 46) "without assuming prior knowledge about anomalies". I find that misleading, prior are always needed in order to generalize anything to an unseen test. Even more so in unsupervised settings. In this case, the priors are the inductive biases induced by the augmentations among other things.
>
> We agree with the reviewer that the referred statement can be misleading, and we always leverage prior knowledge to some extent. We want to emphasize that we only leverage prior knowledge about *normal* samples when designing the objective function. The same applies to the design of context augmentations, which model the distinctiveness and alignment of *normal* samples. This differs from previous works that try to model or incorporate knowledge about anomalies in their objective functions. We updated the manuscript in the following way:
>
>
> *Unlike previous works, which focus on prior knowledge about anomalies, the proposed Con2 models properties of normal samples, which is particularly useful in more specialized data, such as in the medical domain, which we demonstrate in our experiments.*
>
> ---
>
> > I found Figure.1 more confusing than helpful.
>
> Thanks for this comment. We always seek to improve the individual parts of our submission(s). However, we would appreciate some more specific details about what could be improved in Figure 1.
>
> ---
> > Please compare to more pre-trained-based methods. E.g., (Cohen & Avidan, 2022; Reiss & Hoshen, 2023; Li et al., 2023) cited by your paper.
> > Please extend this comparison to more medical datasets. E.g., ChestX-ray14, HAM10000.
>
> Adding more datasets and benchmark results is always valuable. However, we strongly believe that the chosen set of baseline methods and datasets is valuable and shows the advantages of the proposed method. In addition, implementing and running more experiments is not feasible in the short amount of time during the rebuttal, given our computing resources.
>
> ---
>
> Thank you again for your feedback. We welcome any additional suggestions, questions, or requests for information and encourage further discussions.

---

> ### Comment · Reviewer_ENmc · 2024-11-24
>
> I thank the authors for their response.
>
> ***“Exploring pre-trained models in more specialized areas is an interesting direction for future work.”***
>
> I understand that’s not the focus of your work, but (i) I think the manuscript is still misleading in classifying all methods that use pre-training as "Outlier Exposure" (ii) I am still not convinced that “pure SSL” method outperform other methods, even on the evaluated dataset (they might, but the paper does not show it convincingly in my opinion)
>
> ***“We want to emphasize that we only leverage prior knowledge about normal samples when designing the objective function”***
>
> Respectfully, that's still not accurate. Prior knowledge also includes inductive bias - some assumptions regarding unseen samples. Prior knowledge must be used even in supervised classification (e.g. “No Free Lunch” theorems), and even more so in unsupervised settings. I do agree that the wording in the revised version is not as problematic.
>
> ***“We always seek to improve the individual parts of our submission(s). However, we would appreciate some more specific details about what could be improved in Figure 1.”***
>
> A few options are:
> (i) Color the arrows in more distinct color
> (ii) Remove the figure
> (iii) Keep it if other readers find it useful
>
> In any case, that’s not a major concern for me. Just trying to help :-)
>
> ***“we strongly believe that the chosen set of baseline methods and datasets is valuable and shows the advantages of the proposed method.”***
>
> I respectfully disagree that it shows that. A main claim by the paper (the advantage on SSL methods over pretrained ones in specific cases) is supported with the evaluation of a single pretrained method, on only two dataset
>
> Therefore, I remain with my initial score.

---

> ### Author Response · Authors · 2024-11-25
>
> Thank you for your thoughtful feedback. We respectfully disagree with some of the comments made and provide additional clarifications below.
>
>
> **Outlier Exposure and Pretrained Models**
>
> As defined in [9], Outlier Exposure (OE) involves external datasets and pretrained models inherently fall under this category since they are trained on such datasets. Additionally, the superiority of contrastive SSL AD methods over other SSL approaches is well-documented in [1,2,5].
>
>
> **Prior Knowledge and Assumptions**
>
> We do not introduce additional assumptions about anomalies beyond those already made for normal samples. Could the reviewer clarify where our reasoning may diverge from this?
>
>
> **Main Claims of the Paper**
>
> Our main claim is not about comparing SSL methods to pretrained models but rather about the advantage of leveraging prior knowledge of normal samples over simulating anomalies, which is the current state-of-the-art for contrastive SSL-based AD [1,2]. We believe the evaluations included are sufficient to support this claim.
>
>
> We hope these clarifications address your concerns and thank you again for your feedback.
>
>
> [9] Hendrycks, Dan, Mantas Mazeika, and Thomas Dietterich. "Deep anomaly detection with outlier exposure." arXiv preprint arXiv:1812.04606 (2018).

---

> ### Comment · Reviewer_ENmc · 2024-11-28
>
> **Outlier Exposure and Pretrained Models**
>
> While both Pretrained Models and other Outlier Exposure use pre-trained data, they succeed or fail in different situations. Therefore, I believe that while the terminology may differ between papers, the distinction is important.
> I am not sure to which paper list the numbers [1,2,5] refer to, but to the best of my knowledge, pretrain models outperform in most cases.
>
> **Prior Knowledge and Assumptions**
>
> The claim about the use of prior knowledge is not specific to your method. All methods that aim to generalize must rely on some prior assumptions. The original paper's language was confusing; I agree the new language is not as problematic.
>
> **Main Claims of the Paper**
>
> 1. I think that if that’s the paper's main claim, the focus on SSL should be better motivated.
> 2. Comparing a single non-pure-SSL dataset on only two datasets to claim it outperforms is even more problematic than stating you choose to focus on SSL for another reason.

---

> > ### Author Response · Authors · 2024-11-30
> >
> > Thank you for your thoughtful feedback. We value your perspective on the differences between outlier exposure and pretrained models. Could you kindly point us to references illustrating the failure cases of these approaches or clarify this further? We will ensure this distinction is more explicit in the next iteration and note that the references [1, 2, 5] are detailed in the general response.
> >
> > We also appreciate your acknowledgment that the revised formulation clarifies the use of prior knowledge. Regarding the paper's main claims, we will improve the motivation for focusing on SSL and address dataset comparison concerns when revising our work. We appreciate your constructive remarks and are happy to address any additional concerns you may have.

---

> > > ### Comment · Reviewer_ENmc · 2024-12-03
> > >
> > > Thank you for your response and your kind words.
> > >
> > > ***"Could you kindly point us to references illustrating the failure cases of these approaches or clarify this further"***
> > >
> > > See for example the discussion and evaluation of pertaining and OE relative strengths (Tab.1 in [1]). Broadly, OE works well when both outliers and the anomalies are "additional classes" of the normal distribution; and not so well when the outliers are well-separated from the normal samples (but the anomalies are close).
> > >
> > > Though I cannot increase the score for the current submission; I am not opposed to it if other reviewers are more positive, and I wish the authors all the best in their future work
> > >
> > > [1] Reiss, Tal, et al. "Panda: Adapting pretrained features for anomaly detection and segmentation." Proceedings of the IEEE/CVF Conference on Computer Vision and Pattern Recognition. 2021.

---

> > > > ### Author Response · Authors · 2024-12-03
> > > >
> > > > Thank you for your detailed response and the fruitful discussion. While we regret to hear that you are unable to increase the score, we appreciate your thoughtful feedback and plan to incorporate your insights into our work in the future.

---

### Official Review · Reviewer_tEZB · 2024-11-02

**Soundness:** 2
**Presentation:** 2
**Contribution:** 2
**Rating:** 5
**Confidence:** 3

**Summary:**

Dear Authors, thanks for submitting to ICLR.

In this paper, the authors propose a novel anomaly detection method with context-aware data augmentation, termed $CON_2$. Given only normal training data, $CON_2$ augments the dataset through a generator, producing a diverse set of synthetic instances that retain similarity to the original data while introducing unique variations. This generator is designed using an AutoEncoder structure enhanced by contrastive learning and a specialized loss function.

The anomaly detection component in $CON_2$ utilizes two primary metrics: (1) it assesses the presence of similar instances to the input data, incorporating the generator’s output, and (2) it applies data augmentation to the input at test time, subsequently measuring the average similarity between the real and generated data. Evaluation on representative datasets demonstrates that $CON_2$ outperforms state-of-the-art methods in certain cases when optimally configured.

**Strengths:**

1. The proposed method mitigates bias introduced by synthetic anomalies by exclusively leveraging augmented normal data for anomaly detection. Rather than relying on artificially generated anomalies, it detects anomalies by augmenting the normal dataset and comparing the input against an expanded set of potential normal instances, including both real and synthetic normal data.
2. This algorithm requires no anomalous data during training, a practical approach given the rarity and unpredictability of anomalies.
3. Leveraging AutoEncoder and contrastive learning, the model effectively extracts critical information and represents it from multiple perspectives, enabling it to discern whether anomalies arise from content, context, or both.

**Weaknesses:**

Technical Aspect:
1. The method is sensitive to context augmentation methods, which is demonstrated in evaluation part as well. However, there is no systematic method about how to choose proper augmentation method. If use wrong augmentation, this method may perform worse than baselines. (see Table 1). Suggestion: you may consider using validation data to automatically choose augmentations, or if you could provide guidelines for selecting augmentations based on dataset characteristics.
2. Even consider optimal augmentation method only, the performance gain to baselines is marginal in many cases. For example, I cannot tell $CON_2$ is better than baselines in Figure. 4. For instance in Figure 4, UniCon-HA outperform $CON_2$ on Bird, Deer, Horse, and Ship, 4 out of 11 classes. I would suggest statistical tests to demonstrate the significance of their performance gains over baselines, such as t-Test. It would also be helpful to measure the epistemic uncertainty by runing multiple times with different random seed, so that you will know if the the 7 out 11 performance gain is the worst case or best case.
3. The second anomaly score is questionable, start from line 306, page 6. With augmentation, we should obtain different representations of the input data, some of them maybe aligned with existing normal data well. Therefore, we should use some metric like maximal score to highlight the matching, instead of averaging out the potential matching (might be sparse). Suggestion: you might compare the current approach with a max-based score and discuss the trade-offs between the two approaches.
4. Ablation study is not comprehensive. It cannot show how the loss function influence the performance, and does not mention how the anomaly detection metric influence the detection result. As a result, the loss function and metrics are not properly reasoned and tested. Suggestion: please add the experiment with solely one of two loss functions, and also add the result the anomaly detection result with just one anomaly score as metric.

Writing:
1. Figure 1 is not referred at the beginning of the paper but it is put to page 2, and it is hard to understand. The loss function is introduced on page 5, section 3.2, but it is mentioned here without any explanation or reference. To address this, the author may refer to figure 1 in the later part of introduction, when introducing the proposed solutions. Please add a cross reference to the definition of loss function in the caption of Figure 1.
2. Figure 2 and Figure 3 are not on the page they are refered to. Please nsure all figures are properly placed and referenced in the final version.
3. Page 6, line 311, the anomaly score function have a new symbol at the last component. I would suggest a clarification in the following text right after the equation to explain what the "$\circ$" means.
4. Please add number to the equations.

**Questions:**

1. Please correct or clarify the writing issues in the weaknesses.
2. Can you please clarify what the "$\circ$" means in page 6 line 311?
3. Can you please explain the second metric for anomaly score? Why take the average 1/A and why sum between 1~A/2 interval?
4. Please clarify why the average is used in the second anomaly score? From the definition, I think a maximal value based metric makes more sense. Please add this comparision in section 4, or around the ablation study.
5. Can you please add the following ablation study: (1) using L_{context} or L_{content} only, (2) show how each of the two anomaly score contribute to the anomaly detection decision.
6. In the evaluation, please add more details about the result, including a breakdown of precision, recall, and F1 score.
7. Can you please clarify if we can select optimal augmentation method automatically? Otherwise, it may not able to guarantee the performance gain.

**Details Of Ethics Concerns:**

No such concern

---

> ### Author Response · Authors · 2024-11-19
>
> Dear reviewer tEZB,
>
> We would like to thank you again for your detailed review. Below, we answer your questions and address your concerns.
>
> > The method is sensitive to context augmentation methods, which is demonstrated in evaluation part as well. However, there is no systematic method about how to choose proper augmentation method. If use wrong augmentation, this method may perform worse than baselines. (see Table 1). Suggestion: you may consider using validation data to automatically choose augmentations, or if you could provide guidelines for selecting augmentations based on dataset characteristics.
>
> We thank the reviewer for their remark. The two properties introduced in our manuscript, *distinctiveness* and *alignment*, serve as guidelines for selecting augmentations tailored to the characteristics of the dataset. Further, we can compare context augmentations by monitoring the validation loss on a separate set of normal samples and choosing the augmentation that results in the lowest validation loss.
>
> ---
>
> > Even consider optimal augmentation method only, the performance gain to baselines is marginal in many cases. For example, I cannot tell Con2 is better than baselines in Figure. 4. For instance in Figure 4, UniCon-HA outperform Con2 on Bird, Deer, Horse, and Ship, 4 out of 11 classes. I would suggest statistical tests to demonstrate the significance of their performance gains over baselines, such as t-Test. It would also be helpful to measure the epistemic uncertainty by runing multiple times with different random seed, so that you will know if the the 7 out 11 performance gain is the worst case or best case.
>
> In our experiments with natural image datasets, we demonstrate that our approach matches or surpasses the performance of previous contrastive methods on these benchmarks. Additionally, we highlight its clear advantages on specialized medical datasets, where generating synthetic anomalous samples is more challenging. Considering the large body of excellent research already conducted in anomaly detection, these results are very promising.
>
> ---
>
> > Figure 1 is not referred at the beginning of the paper but it is put to page 2, and it is hard to understand. The loss function is introduced on page 5, section 3.2, but it is mentioned here without any explanation or reference. To address this, the author may refer to figure 1 in the later part of introduction, when introducing the proposed solutions. Please add a cross reference to the definition of loss function in the caption of Figure 1.
>
> > Figure 2 and Figure 3 are not on the page they are refered to. Please nsure all figures are properly placed and referenced in the final version.
>
> Thank you for spotting this. We added more cross-references to the figures and the loss functions. However, we believe it to overly strict to force all tables and figures to be on the same page as where they are mentioned for the first time.
>
> ---
>
> > The second anomaly score is questionable, start from line 306, page 6. With augmentation, we should obtain different representations of the input data, some of them maybe aligned with existing normal data well. Therefore, we should use some metric like maximal score to highlight the matching, instead of averaging out the potential matching (might be sparse). Suggestion: you might compare the current approach with a max-based score and discuss the trade-offs between the two approaches.
>
> > Please clarify why the average is used in the second anomaly score? From the definition, I think a maximal value based metric makes more sense. Please add this comparision in section 4, or around the ablation study
>
> > Can you please clarify what the "o" means in page 6 line 311?
>
> > Can you please explain the second metric for anomaly score? Why take the average 1/A and why sum between 1~A/2 interval?
>
> We follow previous work [1, 2] by using the average score over test set augmentations, and note that this is generally the standard when using test-time augmentations [3].
>
> We take the sum over $A/2$ values as there are two sums inside the brackets and both sum over $A/2$ values, which in total sums over $A$ terms. Hence, we also scale the loss with $1/A$.
>
> The symbol $\\circ$ defines a composition of functions, see [Wikipedia, Function Composition](https://en.wikipedia.org/wiki/Function_composition) or [8], and
> $ t_i \\circ t_{\\mathcal{C}}= t_i(t_{\\mathcal{C}}(\\cdot)) $
> .
>
> ---

---

> > ### Author Response · Authors · 2024-11-19
> >
> > > Ablation study is not comprehensive. It cannot show how the loss function influence the performance, and does not mention how the anomaly detection metric influence the detection result. As a result, the loss function and metrics are not properly reasoned and tested. Suggestion: please add the experiment with solely one of two loss functions, and also add the result the anomaly detection result with just one anomaly score as metric.
> > > Can you please add the following ablation study: (1) using L_{context} or L_{content} only, (2) show how each of the two anomaly score contribute to the anomaly detection decision.
> >
> > We performed an additional ablation study, where we analyzed the anomaly detection performance with only one of the two losses. In the following table, we can see the results for the individual loss terms:
> > | Method                   | Score                | Pneumonia         | Melanoma          |
> > |--------------------------|----------------------|--------------------|--------------------|
> > | $\mathcal{L}_{\text{Content}}$ | $\mathcal{S}_{\text{LH}}$ | $92.3 \pm 0.9$    | $92.8 \pm 0.2$    |
> > | $\mathcal{L}_{\text{Context}}$ | $\mathcal{S}_{\text{LH}}$ | $79.9 \pm 1.3$    | $92.7 \pm 0.5$    |
> > | $\mathcal{L}_{\text{Con}_2}$   | $\mathcal{S}_{\text{LH}}$ | **$93.0 \pm 0.3$** | **$94.0 \pm 0.3$** |
> > | $\mathcal{L}_{\text{Content}}$ | $\mathcal{S}_{\text{NND}}$ | $89.6 \pm 0.4$    | $93.1 \pm 0.3$    |
> > | $\mathcal{L}_{\text{Context}}$ |  $\mathcal{S}_{\text{NND}}$ | $81.4 \pm 1.6$    | $92.5 \pm 0.8$    |
> > | $\mathcal{L}_{\text{Con}_2}$   | $\mathcal{S}_{\text{NND}}$ | **$93.9 \pm 0.3$** | **$94.5 \pm 0.2$** |
> >
> >
> > We see that having the combination of context and content loss performs the best across the two datasets and anomaly score functions. It is interesting to see that the context loss consistently improves the performance compared to the content-only objective.
> > We also added the additional results to Appendix E.3.
> >
> > ---
> >
> > > In the evaluation, please add more details about the result, including a breakdown of precision, recall, and F1 score.
> >
> > We follow previous works regarding the choice of metrics for reporting the anomaly detection performance [1,2]. We will add more scores in the appendix of the camera-ready version.
> >
> > ---
> >
> > > Can you please clarify if we can select optimal augmentation method automatically? Otherwise, it may not able to guarantee the performance gain.
> >
> > It is possible to select the best-performing context augmentation based on a validation set by tracking the validation loss. In this work, we focused on highlighting the different performances coming from different context augmentations (Flip, Invert, Equalize) and how they relate to the properties defining a valid context augmentation, *distinctiveness*, and *alignment*.
> >
> > ---
> >
> > Thank you again for your feedback. We welcome any additional suggestions, questions, or requests for information and encourage further discussions.

---

> > > ### Author Response · Authors · 2024-11-28
> > > **Reminder**
> > >
> > > Dear Reviewer tEZB,
> > >
> > > Thank you for your thoughtful feedback. We have carefully addressed your comments and concerns, and we hope our responses have clarified the strengths and contributions of our work. If there are any remaining questions or points of discussion, we would be happy to provide further clarification.

---

> > ### Comment · Reviewer_tEZB · 2024-11-29
> >
> > Dear Authors,
> >
> > Thanks for your timely reply, here are some further questions for me to understand the advantage of your method.
> >
> > > "...we highlight its clear advantages on specialized medical datasets, where generating synthetic anomalous samples is more challenging. "
> >
> > Could you provide more insights into **why medical datasets are more challenging** and how CON2 addresses these challenges (while the baselines fail)? Additionally, how can we determine when to use CON2 effectively?

---

> > > ### Author Response · Authors · 2024-11-30
> > >
> > > Dear Reviewer tEZB, thank you for your thoughtful follow-up questions and for engaging with our work.
> > >
> > > Medical datasets are uniquely challenging because we typically lack sufficient knowledge about anomalies to simulate them effectively, a common approach in prior works [1,2]. Additionally, foundation model-based methods like CLIP-AD have shown underwhelming performance in this domain. CON2 addresses these challenges by leveraging context augmentations that only require *alignment* and *distinctiveness*—properties derived solely from the normal data distribution. For instance, in pneumonia detection, histogram equalization preserves information (*alignment*) while providing a distinct way of looking at the data (*distinctiveness*).
> > >
> > > As demonstrated in Sections 4.1 and 4.2, CON2 performs well across diverse settings but is particularly advantageous when assumptions about anomalies are limited or undesired. We hope this clarifies your questions and are happy to address any additional concerns you may have.

---

> > > > ### Comment · Reviewer_tEZB · 2024-12-02
> > > >
> > > > Dear Authors,
> > > >
> > > >
> > > > Thank you for your timely clarification.
> > > >
> > > > From what I observe in Section 4.1, particularly Figures 4 and 5, there doesn’t appear to be a consistent performance improvement. While the results on medical datasets show promise, they require further evidence and stronger justification to be fully convincing.
> > > >
> > > > Overall, the current version of this method does not outperform the baseline on standard tasks. Although it demonstrates strength on specific medical datasets, the reasoning behind the performance gains remains insufficient. I recommend additional effort to improve both overall performance and the analysis of the observed gains. Therefore, I will maintain my current score.
> > > >
> > > > Thanks

---

### Author Response · Authors · 2024-11-19
**Reply to all Reviewers**

Dear reviewers,

We would like to thank all reviewers for providing comprehensive and valuable feedback. We particularly value the reviewers’ recognition of our strong results [**ENmc**], especially on medical data [**h61g**, **yEuR**], applicability and ease of use of the proposed method [**tEZB**, **yEuR**], and the good writing and illustrations [**ENmc**, **yEuR**].

Your thorough reviews have helped us identify potential areas for improvement and gain valuable perspectives that strengthen the contribution of our research. We have carefully addressed your comments. We included additional results and clarified some points in the updated manuscript based on the feedback [**tEZB**, **ENmc**, **h61g**, **yEuR**].

We added an ablation study evaluating the effect of the individual loss components as suggested by reviewers **h61g** and **tEZB**. In addition, we re-run the experiments on the melanoma dataset to increase the consistency across datasets between NND and LH anomaly scores [**h61g**].
In the manuscript, we added additional cross-references and equation numbers to improve readability [**tEZB**], we changed the wording about the use of prior knowledge [**ENmc**], and we added some explanation regarding the difference between context and content augmentations [**yEuR**].


We welcome any additional suggestions, questions, or requests for information and encourage further discussions.
Once again, we thank the reviewers for their time and effort in evaluating our work.

---

### References


[1] Tack et al., “CSI: Novelty Detection via Contrastive Learning on Distributionally Shifted Instances”, Neurips 2020

[2] Wang et al., “Unilaterally Aggregated Contrastive Learning with Hierarchical Augmentation for Anomaly Detection”, ICCV 2023

[3] Krizhevsky, Alex, Ilya Sutskever, and Geoffrey E. Hinton. "Imagenet classification with deep convolutional neural networks." Neurips 2012.

[4] Ruff, Lukas, et al. "Deep one-class classification." ICML 2018.

[5] Bergman, Liron, Niv Cohen, and Yedid Hoshen. "Deep nearest neighbor anomaly detection." arXiv 2020.

[6] Sun, Yiyou, et al. "Out-of-distribution detection with deep nearest neighbors." ICML 2022.

[7] Ruff, Lukas, et al. "A unifying review of deep and shallow anomaly detection." Proceedings of the IEEE 109.5 (2021): 756-795.

[8] Bourbakil, “Fonctions d'une variable réelle”, 1949

---

### Meta-Review · Area_Chair_sJLs · 2024-12-14

**Metareview:**

The idea sounds interesting (at least to me), though our reviewers think its novelty is not enough. For example, what is the difference between context transformations and ordinary transformations given there are already contrastive anomaly detection methods? Another major issue is clarity, for example, why the context-specific cluster structure is a good way to learn compact and informative representations? How did you come up with the goals "distinct" and "aligned" and why they are helpful for the purpose? They should definitely be well explained in the introduction (or even the abstract) level, because they are so related to the idea that without knowing them we don't know either why the proposed method works and when it would not work.

**Additional Comments On Reviewer Discussion:**

The rebuttal didn't well address the concerns from the reviewers.

---

### Decision · Program_Chairs · 2025-01-22

Reject